# VISUAL SCRATCHPADS: ENABLING GLOBAL VISUAL REASONING

## ABSTRACT

Modern vision models have achieved remarkable success in benchmarks where a small subset of local features provides critical information about the target. There is now a growing interest in solving tasks that require more global reasoning, where local features offer no significant information. These tasks are reminiscent of the connectivity problems discussed by Minsky and Papert in 1969, which exposed the limitations of the perceptron model and contributed to the first AI winter. In this paper, we revisit such tasks by introducing four global visual benchmarks involving path findings and mazes. We show the following: (1) Although today's large vision models largely surpass the expressivity limitations of the early models, they still struggle with learning efficiency; we introduce the 'globality degree' to understand this; (2) we then demonstrate that the outcome changes and global reasoning becomes feasible with the introduction of a 'visual scratchpad'; similarly to the text scratchpads and chain-of-thoughts used in language models, visual scratchpads help break down global problems into simpler subproblems; (3) we further show that more specific 'inductive scratchpads', which take steps relying on less information, afford better out-of-distribution generalization and succeed for smaller model sizes.

## 1 INTRODUCTION

Modern computer vision models, as well as text models, are often pre-trained on vast datasets encompassing much of the knowledge available on the internet. While this has led to impressive capabilities, there is growing concern that these models may make decisions based on shallow, local information rather than engaging in deep, complex reasoning. Evidence suggests that many of these models function primarily through retrieval, acting as blurry, compressed versions of the Internet. These models excel at smooth interpolation within this encoded knowledge but often fail to grasp the underlying logic and complexity of the real world. Unfortunately, the community lacks benchmarks that rigorously test a model's ability to perform global reasoning and multi-step problem-solving in the visual domain. Instead, most common visual benchmarks are limited to tasks that can be solved with superficial cues and local features. In this work, we aim to close this gap by exploring whether current models are capable of learning tasks that require deep, multi-step global processing.

In order to address this, it is crucial to define the characteristics of global visual tasks. In contrast to local tasks, where a small subset of pixels—typically organized into patches—is sufficient to achieve better-than-random accuracy, global tasks require a more holistic understanding of the entire visual scene. For example, in ImageNet classification (Deng et al., 2009), a single patch containing cat whiskers significantly increases the likelihood that the model will classify the image as a cat. This reliance on local features is further exemplified by the effectiveness of drastic image cropping in object-centric datasets, where self-supervised models such as DINO (Caron et al., 2021) employ aggressive multi-crop strategies, sometimes cropping as much as 90% of the image which empirically improves the performance. Humans, in contrast, do not rely solely on local information; for instance, when driving a car, it is insufficient to focus only on the view directly in front of the vehicle. A competent driver must recognize multiple visual objects in the environment and consider their complex behaviors before making decisions. Yet, using such complex real-world tasks, like autonomous driving, to study model learning is impractical due to their complexity and unpredictability. Instead, we need interpretable and deterministic tasks with well-defined data generation processes to assess the reasoning ability of the models.

To address this need for visual tasks with global multi-step reasoning, we propose four simple datasets, some reminiscent of connectivity task Minsky & Papert (1969) that played a significant role in the AI winter (see Figure 14). Our tasks are closely related to these early studies as they require an understanding of the connectivity concept. In particular, we propose a graph connectivity task, the task of determining the number of strings in an image, and a maze solvability task where we consider rectangular and circular mazes. We argue that these tasks possess the key ingredients for testing global reasoning. They are inherently global because understanding a small portion of the graph or maze offers no meaningful insight into the final label (whether the structure is connected or not). At the same time, their data generation processes are fully controllable and deterministic, allowing for straightforward manipulation of task complexity by simply increasing the number of nodes in the graph or cells in the maze. We have also developed more visually engaging variants, namely the strings task and the circular mazes to enhance the visual aspect. These datasets enable us to simultaneously test both reasoning and visual recognition abilities, which is the core objective of this paper.

Despite the increased expressivity of modern vision models compared to the perceptrons discussed by Minsky & Papert (1969), current models still struggle with global tasks. While they can solve simple and small graph connectivity and maze problems, their performance rapidly declines to the accuracy of a random model as the tasks become more complex. This deterioration occurs regardless of the model's size, the task used, or whether a pre-trained checkpoint is employed (see Section 4.1). To remedy this issue, we put forward the notion of *visual scratchpads*. Similar to the scratchpad idea used in the text domain (Nye et al., 2021), a visual scratchpad is a single frame or a sequence of frames that depict the underlying reasoning behind the label of a sample, e.g., the existence of a path between the source and sink nodes in a maze. The model is supervised with the scratchpads during training and has to generate them at test time. The scratchpad acts as a guide, showing the model how to decompose the global problem into simpler subproblems, such as coloring two nodes at a time in a graph or a few cells at a time in a maze. Interestingly, we find that even using a one-shot single-frame scratchpad, which only provides a visualization of the final solution boosts performance significantly making the model capable of learning most of the considered tasks. Furthermore, the model exhibits a hierarchical "staircase" learning behavior, learning the solution incrementally during training, even though it was trained to generate only the final solution in a single shot. Moreover, for the multi-frame scratchpad, we propose a model that generates scratchpad frames in an autoregressive, recurrent, and Markovian manner called the inductive scratchpad model. We show that this model outperforms the single-scratchpad model both in in-distribution and out-of-distribution settings thanks to its Markovian modeling and adaptive compute time at inference.

Our work differs from previous efforts in textual scratchpads due to the unique characteristics of visual data. Unlike language, which consists of discrete tokens, vision deals with continuous inputs. Additionally, vision operates within a 2D spatial neighborhood of pixels or patches, in contrast to the linear, 1D structure of text. This difference influences how models generalize to OOD samples (such as generalization to more challenging samples that require more reasoning steps), as more objects can be represented within the same pixel space without requiring additional positional embeddings. These distinct features make vision a particularly interesting and fertile field for applying and extending the scratchpad concept. Here is a summary of this paper's contributions:

- **Exploration of locality and globality in the visual domain**: we analyze the concept of locality/globality in vision, to distinguish between global and local tasks while paying particular attention to the vision Transformers (ViTs, Dosovitskiy et al., 2020), the currently dominant models.

- **Development of datasets inspired by Minsky & Papert's connectivity task**: we revisit the foundational work of Minsky & Papert (1969) by proposing four tasks related to the connectivity concept that require global reasoning and are hard to learn for ViT models of different sizes.

- **Introduction of visual scratchpads for global reasoning**: we introduce the visual scratchpad to enable multi-step reasoning in vision models. More specifically,

  - we show that a single-frame scratchpad model can learn visual tasks that were not learnable with the no-scratchpad models irrespective of size and pre-training;
  - we introduce a recurrent model for generating multi-frame scratchpads, namely, the inductive scratchpad model that allows for better in-distribution and out-of-distribution (OOD) generalization thanks to its Markovian modeling and adaptive compute time at inference.

## 2 GLOBAL VISUAL REASONING DATASETS

Vision models have shown remarkable performance on different tasks including image classification, image segmentation, object detection, etc. However, these mainstream visual tasks have two characteristics in common:

1. Local features in the image are informative. For example, if we consider an image partitioned into a set of patches, there is usually a small subset of patches that provides significant information on the target (e.g., the label).

2. These tasks can be instinctively and instantaneously solved by humans. That is humans do not need to ponder for longer periods of time to solve these tasks. Considering the System 1 / System 2 terminology (Kahneman, 2011), these visual tasks are dealt with by our System 1. In general little or no multi-step chain of entailments is necessary to solve these tasks (e.g., no search).

Despite being common, not all visual tasks share these characteristics. As an example, consider solving a maze, i.e., answering whether two points are connected in a maze or not. Assuming the size of the maze is large enough, humans need some deliberation before solving the maze. Normally, humans would follow paths with a pen on the maze to see where the starting point leads to. Importantly, apart from trivial edge cases where the start and end locations are close, local features are not informative on the maze task. For instance, if only three patches of a maze are given, one cannot solve the maze (determine whether there is a connection) with high probability. Motivated by the latter, we propose the following visual datasets in this paper:

• **Connectivity datasets.** Inspired by Minsky & Papert (1969), we consider two datasets based on the notion of connectivity.

  – **Cycles task.** In this task, $2n$ nodes are drawn randomly (on an invisible circle) in the image. There are also $2n$ edges between these nodes that form either one cycle of size $2n$ or two cycles of size $n$. The task is to determine whether the graph is connected (one cycle, label 1) or not (two cycles, label 0) given an input image. See Figure 1 for an example. In this task, one has reason on at least $n$ nodes and the connections between them to determine the label correctly as any $n-1$ nodes do not provide any information on whether there are two cycles or one. Thus, one can simply increase the complexity of this task by increasing $n$.

  – **Strings task.** In order to further increase the visual complexity, we consider a dataset consisting of random strings. In each sample, there are either two closed strings or one longer closed string. The dataset generation process for these curves is similar to the cycles task above with the difference that in the strings we do not make the (anchor) nodes visible and also connect them using 3rd-degree Bézier curves which produces continuous strings (see Figure 1). Similar to the cycles task, one can increase the complexity of this task by increasing the number of invisible anchor points $2n$ which leads to longer more entangled strings.

• **Maze solvability.** We also consider a maze task in which there are always two connected components, and we have a start/source point (shown in blue) and an end/sink point (shown in red). The source and sink are in the same connected component or not equiprobably. The task is to determine whether they are connected (label 1) or not (label 0). We provide this dataset in a rectangular and a circular version to increase the visual complexity. Examples can be seen in Figure 1. To adjust the complexity of maze datasets, one can modify the size of the maze and hence the number of cells, size of the components, and distance between the source and sink (if connected).

For each task, there exists a natural visual scratchpad that uncovers the underlying reasoning behind the label. For the maze, similar to what humans do we can start coloring from the source cell (e.g., the cell in blue) to see which areas are reachable until reaching the sink cell or the end of the maze region similar to doing a breadth-first search (BFS). This coloring is similar to what humans would naturally do by following the paths from the beginning to see which one (if any) leads to the sink cell. For the cycles task, we can use a similar idea, we can start by coloring one node, and then coloring all of the nodes that are connected to this node (which is either half of the graph or all of the graph). Analogously, for the strings task the visual scratchpad would be coloring one of the strings if there are two strings or coloring the whole string if there is only one. To disambiguate which cycle/string to color, we always color the cycle/string that passes through the rightmost (anchor) node.

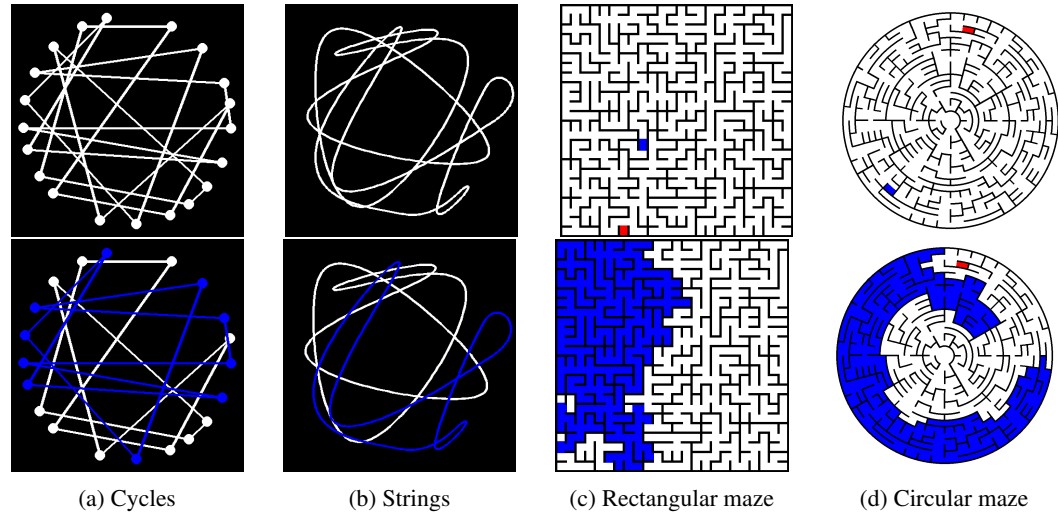

| (a) Cycles | (b) Strings | (c) Rectangular maze | (d) Circular maze |

Figure 1: Examples of different tasks. The first row shows the inputs. The second row shows the complete scratchpad (e.g., the target frame in the single-frame scratchpad model and the final frame in the multi-frame scratchpad). In the examples of cycles and strings, we have two different cycles and strings. In the example of the circular maze, the two cells are not connected while in the example of the rectangular maze, they are. In each of the scratchpads, we start coloring from some point until the whole connected part is colored (or the sink cell is reached in the maze examples).

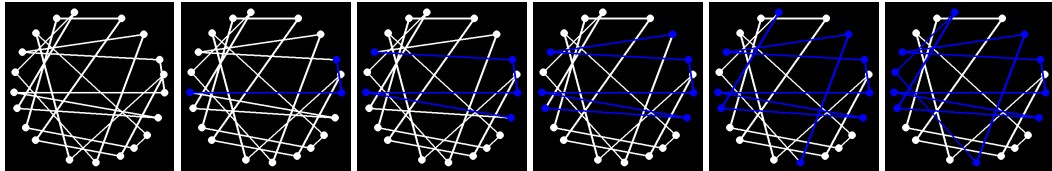

Figure 2: Example of the cycles task showing how the scratchpad can contain several frames. The input image is presented on the left side and then different frames of the visual scratchpad are depicted from left to right ending with the complete image.

Note that the scratchpad can have a single frame format in which the full scratchpad (all coloring done) is shown. The scratchpad can also be generated in multiple frames, i.e., consecutive frames that lead to the final frame. This scratchpad is again analogous to what humans do: starting from one point and coloring progressively. For example, this could be coloring a distance of $10$ when doing the search for the maze problems, and (up to) two (anchor) nodes for the cycles and strings tasks. An example of doing so for cycles task is depicted in Figure 2. We start the coloring from the source node in mazes and from the rightmost (anchor) node in cycles and strings examples. See Appendix F for scratchpad figures for other datasets.

## 2.1 GLOBALITY, LOCALITY AND SCRATCHPADS IN THE VISUAL DOMAIN

Here, we further discuss the meaning of locality and globality in the visual domain and the connection of the visual scratchpad to the concept of scratchpad in the text domain (Nye et al., 2021).

Recently, Abbe et al. (2024) proposed the notion of globality degree to explain why some tasks are hard to learn for Transformers and also to explain the effectiveness of scratchpads in the textual domain. Considering input tokens $X_1, \ldots, X_n$ and output $Y$, the globality degree of a task is defined as the minimum number of tokens $k$ such that there exist $k$ tokens $X_{i_1}, \ldots, X_{i_k}$ that along with the histogram of tokens $\hat{P}_X$ [1] provide significant information on the target $Y$, i.e., $I(X_{i_1}, \ldots, X_{i_k}, \hat{P}_X; Y) = n^{-O_n(1)}$ where $I$ is the mutual information. It is further conjectured, with empirical support, that the learning complexity of tasks increases with their globality degree,

---

[1] In the textual domain, histogram simply refers to reporting how many times each token is appearing regardless of its position (similar to the bag of words).

and Transformers can only learn tasks with a constant globality degree (in $n$) efficiently (polysize model and polymany iterations). We extend this definition to vision tasks learnable by ViTs.

**Definition 1. Globality degree** *(in vision with significant patch regime). Assume images are partitioned into patches $X_1, \ldots, X_n$. We define the globality degree with threshold $\alpha$ of a task as the minimum number $k$ such that there exist patches $X_{i_1}, \ldots, X_{i_k}$ that satisfy $I(X_{i_1}, \ldots, X_{i_k}; Y) \geq \alpha$ where $I$ is the mutual information.*

In words, this is the least number of patches $k^*$ required to obtain an $\alpha$-mutual information with the target. The higher $\alpha$, the more informative these patches are on the target, and the lower $k^*$ (for the same $\alpha$), the less global the task is. One may now try to use this measure to characterize which targets are learnable in polynomial time in the number of patches $n$. This requires a closer investigation of the scale of the parameters. To define our asymptotic quantities properly, we assume that the number of patches $n$ is scaling (e.g., as the size of the maze increases, we need a higher resolution image to solve it).[2] In this case, the requirement would be to have $\alpha = n^{-O_n(1)}$ and $k^* = O_n(1)$ in order for the whole[3] learning complexity to be polynomial in $n$, i.e., we expect the complexity of learning in this regime to depend polynomially on both $n$ and $1/\alpha$ with an exponent given by $k^*$, i.e., $poly(\frac{1}{\alpha}, n)^{k^*}$ where $k^*$ is the globality degree. Our main regime of interest, the regime with significant patches, assumes that patches are large enough, e.g., $P \times P$ sized-patch with $P = \sqrt{n}$, such that only a unique ordering of the patches is valid (if one permutes the patches the new sample does not belong to the distribution's support w.h.p. as in many computer vision tasks of interest). In this case, the histogram part should not be inserted as done in our current definition. This is why we call this the globality degree in the 'significant patch regime', which we consider to be the right regime to better understand the targets of interest. In the small-patch regime where the patches have a constant size, i.e., $P \times P$ sized-patch with $P = O_n(1)$, one would also need to update slightly the definition of the globality degree in order to capture the fact that targets that are permutation invariant may be more easily learnable by the Transformer after dropping the positional embeddings, which results in adding the histogram of the patches to the mutual information as was done in the globality degree for text by Abbe et al. (2024).

Note that a task being "local" by affording a low $k^*$ for a significant $\alpha$ does not mean that it does not depend on all the patches, but that these few $k^*$ patches are sufficient in order to obtain non-trivial information about the target, and this gives the starting point to learning with a significant edge.

According to the definition above, classical vision tasks such as image classification are "local" as a few patches often provide significant information on the class (e.g., having a patch containing a dog's ear). Whereas, in our proposed datasets, seeing a few patches often provides no information on the label. Hence, our proposed datasets have a high globality degree, or in short, are rather "global". For example, in the maze examples, seeing just a few patches from the maze does not help with determining the label. Similarly, for the cycles task of size $2n$, if one only sees the connection between $n-1$ nodes, one cannot have any information on the label.

In order to further support our claim that mainstream vision tasks are local while our proposed tasks are not, we conduct the following experiment. For each sample in a given dataset, we mask the patches with probability $p$ at both training and inference and see the performance of the model with different values of $p$. We perform this experiment for the cycles 12 task and ImageNet. Since the the cycles 12 task is not learnable from scratch we start from a CLIP (Radford et al., 2021) pre-trained ViT-L/14 checkpoint (Fang et al., 2023) for both. The results are shown in Figure 3 (left) [4]. We observe that the model demonstrates good performance on the ImageNet dataset even when 90% of the image is masked while it cannot learn the cycles 12 task once 30% (or more) of the image is masked. The latter shows that the cycles 12 task is a high globality dataset where one needs on average at least 70% of the patches to gain minimal information on the label while ImageNet is a local task where weak learning is possible with only 10% of the patches.

Interestingly, we note that a model trained from scratch was not able to learn the cycles task even when no patch was masked. To better explain this phenomenon, in Figure 4 (right), we compare the

---

[2]The number of patches in ViT models is usually constant as the images are usually resized unless the image is so fine-grained that a higher resolution is required, e.g., when the number of graph nodes diverges.

[3]I.e., assuming networks with polynomial number of edges trained with a polynomial number of samples/epochs, the overall complexity would be polynomial

[4]We use min-max normalized accuracy in the plot, including the random baseline for normalization.

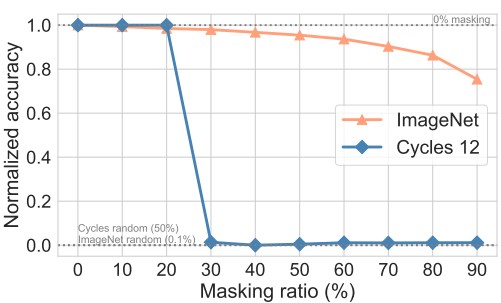 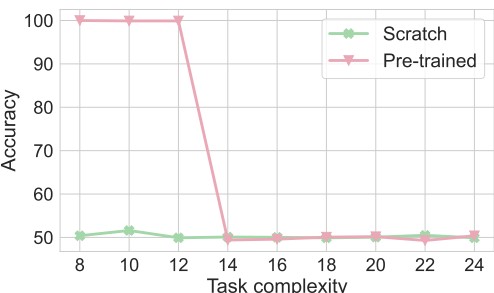

Figure 3: Experimental evidence that Cycles 12 is a more global task w.r.t. common computer vision benchmarks (e.g. ImageNet). Cycles 12 quickly becomes not learnable when more patches are masked, while ImageNet is still far from random accuracy.

Figure 4: Comparison between training from scratch and initializing with a pre-trained model, while varying the task complexity. Pre-training is not sufficient to guarantee convergence for complex tasks. Convergence without pre-training is not possible even for easier tasks.

performance when initializing with a pre-trained model (Fang et al., 2023) versus a model trained from scratch on the cycles task of varying size. This shows that as the task complexity increases with the number of nodes, none of the models are able to learn the cycles task, meaning that even strong internet-sourced priors in the pre-trained model are not helpful.

Next, we discuss the connection of visual scratchpads to scratchpads used in text (Nye et al., 2021) and why it helps. The idea of scratchpad generally refers to training the models with intermediate reasoning steps, so that they generate both the reasoning steps and the final answer during inference. For instance, consider math questions with simple numerical answers. Nye et al. (2021) showed that training language models to first output the intermediate steps of the solution and then the final numerical answer results in superior accuracy than training the model to directly output the answer.

Abbe et al. (2024) have shown that scratchpads can reduce the globality degree and by doing so reduce the learning complexity. More specifically, the scratchpad can provide intermediate targets $Y_1, \ldots, Y_m$ such that $Y_m = Y$ and each $Y_i$ is of low globality given the previous intermediate targets and the input which makes predicting them in a sequential manner easily learnable. The same is true for our proposed datasets and multiple-frame scratchpads as each scratchpad frame is a low globality function of the previous frame and also the label is a low globality function of the final frame. For example, in each frame of the multiple-frame scratchpad for the cycles task at most two nodes are colored, and determining these nodes is possible by using a few patches. Similarly, determining the final label from the final scratchpad frame is a local operation since checking one node's color provides significant information on the label. Note that to compute the label, one has to check whether all nodes are colored or not. However, even checking whether one node is colored or not provides non-trivial information on the output even though it may not determine the output perfectly (if that node is colored). Thus, the multi-frame scratchpad can reduce the learning complexity of tasks by breaking their globality degree. Interestingly, the single-frame scratchpad model can also break the globality degree of the task. As explained before, predicting the label from the complete scratchpad is a local function. Note that learning the full scratchpad from the input image is also a low globality function. To see this, note that coloring the first three nodes of the scratchpad (which corresponds to non-trivial mutual information) is a local function as it requires a limited number of patches which allows weak learning of the scratchpad. A particularly interesting phenomenon is that this weak learning can result in strong learning of the scratchpad frame through hierarchical learning. We explain this in more details in Section 4.4.

## 3 METHODOLOGY

As discussed in the introduction, we have three supervision formats for our visual reasoning datasets: no scratchpad, single-frame scratchpad, and multi-frame scratchpad. We use a vision Transformer (ViT) (Dosovitskiy et al., 2020) backbone for all supervision modes.

**No scratchpad baseline.** In this case, the image is given to the model, and the model has to produce the label directly. We use a ViT architecture with a classification token, CLS, for this setting. We use a linear layer on the CLS token features to compute the label logits. We use the cross-entropy loss function on the logits for training the model. As it will be shown in Section 4.1, this model is not capable of learning the proposed datasets. Hence, we next introduce the single-frame scratchpad.

### 3.1 Single-frame scratchpad

For generating the scratchpad in a single-frame format, we make some modifications to the no-scratchpad architecture above. Here, the model predicts both the label and the scratchpad (the complete one in the single-frame format). We keep the ViT encoder with a CLS as the backbone and add a linear layer to the hidden representation of the last transformer layer to predict the scratchpad image. During training, we use cross-entropy loss to supervise the label and pixel-wise mean-squared loss similar to He et al. (2022); El-Nouby et al. (2024) to supervise the scratchpad image. In Section 4.1, we show that the single-frame model has better performance than the no-scratchpad model, and for large enough models it may be able to learn the proposed tasks. In Section 4.2, we evaluate the out-of-distribution (OOD) performance of this model. In the next part, we introduce our inductive model for multi-frame scratchpads which has superior performance in both the in-distribution and OOD settings.

### 3.2 Multi-frame scratchpad and Inductive scratchpad model

In this section, we introduce the inductive scratchpad model that is used for the multi-frame scratchpad setting where the model predicts all the scratchpad frames in a sequential and Markovian manner. In this setting, the model predicts each scratchpad frame based only on the previous scratchpad frame predicted by the model (or the input at the beginning). For example, the model first predicts the first scratchpad frame using the input image. Then, it uses the first frame to predict the second one, and so on. More precisely, the model has a recurrent component $\mathcal{M}$ that takes an input image (either the input image or a scratchpad frame) as input and predicts three outputs: the next scratchpad frame ($\hat{f}$), the label ($\hat{y}$), and a binary variable for halting ($\hat{h}$). This recurrent module is applied to the input image and the subsequent intermediate frames until the halting signal activates (or an upper limit of recurrences is reached). The predicted label at the last recurrence is the predicted label of the model. Note that generating each scratchpad frame depends only on the last generated frame (or the input image) and there is no part of the recurrent module to record the history. As a result, the model is independent of the number of scratchpad frames used in each sample.

**Training procedure.** During training, we initially used teacher forcing, that consists in providing the model with perfect frames from the training set. However, this approach creates a discrepancy during inference, where the model sees its own generated frames as input. Generated frames may exhibit a slightly different distribution as the reconstructions are not perfectly accurate. While this issue is well studied in text generation, where discrete tokens are used, it becomes more pronounced in vision tasks with continuous outputs. To mitigate this discrepancy, we use an alternated training procedure where the model sees perfect frames 50% of the times, and generated frames the other 50%. This ensures that the model learns to handle imperfect inputs, leading to improved performance during inference. More details on this training procedure can be found in Appendix C.1.

## 4 Experiments

In this section, we show the performance of different methods on our proposed datasets focusing on required model size and OOD generalization. Each of our datasets contains $10^6$ (1M) training samples. See Appendix B for more details on the experiments.

### 4.1 Model size experiments

First, we compare the performance of different methods with varying model sizes on our proposed datasets. In particular, we compare the no scratchpad baseline, the single-frame scratchpad model, and the inductive scratchpad model used for multi-frame scratchpad prediction. Moreover, we use four different sizes for the ViT encoder of our models: small, base, large, and huge, which have

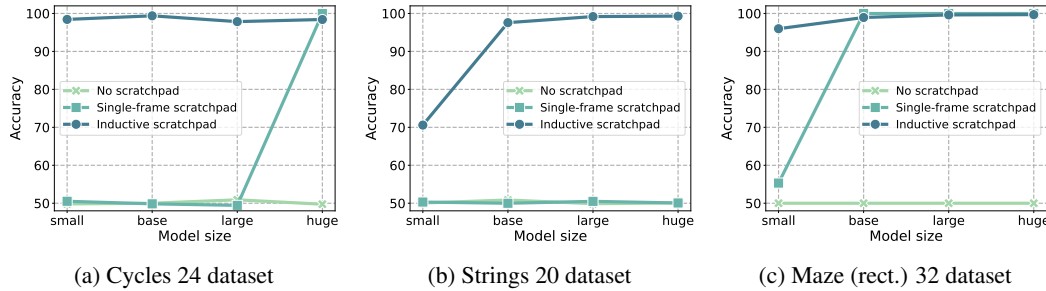

(a) Cycles 24 dataset       (b) Strings 20 dataset       (c) Maze (rect.) 32 dataset

Figure 5: Validation accuracy for different datasets learned by different methods and model sizes. We can see that the model without a scratchpad is not capable of learning any of these tasks, while for large enough models, the single-frame scratchpad model may be able to learn. Further, the inductive scratchpad model can learn all the tasks with smaller models than the single-frame scratchpad model.

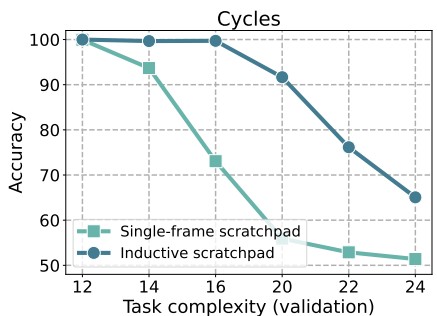

Figure 6: OOD experiments where the model is trained on Cycles 12 and tested on more complex Cycles tasks.

Table 1: OOD performance for maze 24 (rectangular) dataset. While both models are good in-distribution, the inductive scratchpad achieves almost perfect accuracy on OOD while the single-frame model hardly goes beyond the random baseline (50%).

| Method | Accuracy (%) | |
| --- | --- | --- |
| | **ID** | **OOD** |
| Single-frame | **100.0** | 54.4 |
| Inductive | 99.8 | **99.8** |

respectively around $22M$, $86M$, $307M$, and $632M$ parameters (see Appendix C for detailed specifications). The accuracy of different methods with different model sizes is shown in Figure 5. It can be seen that the no-scratchpad baseline is not able to go beyond random accuracy for any of these tasks. On the other hand, the single-frame scratchpad model can learn the cycles 24 and maze (rect.) 32 tasks for large enough models while it still cannot learn the strings 20 task. The inductive scratchpad model used for the multi-frame prediction, however, learns all the proposed tasks even with smaller models. We report the results for the circular maze dataset in Appendix E.

We note that the number of parameters for the three methods is very similar. The single-frame model only adds a linear layer for predicting the scratchpad to the no-scratchpad baseline. Likewise, the inductive scratchpad model only adds a linear layer for predicting the halt signal to the single-frame model. However, during inference, the inductive scratchpad model is applied a variable number of times. Hence, the inductive scratchpad model uses compute adaptively depending on the complexity of the sample, and therefore its inference time compute is usually larger than the baselines.

## 4.2 OOD GENERALIZATION

Next, we consider the out-of-distribution (OOD) generalization performance of different methods where we show the inductive scratchpad model has a superior OOD generalization performance. This observation is due to the fact that the inductive scratchpad model only learns the steps of the reasoning process, and as a result, is independent of the number of reasoning steps required allowing it to generalize to harder problems with its adaptive compute time.

For the cycles task, we can control the complexity of the task with the number of nodes. In particular, for OOD experiments, we consider training on samples with 12 nodes and then testing on samples with a higher number of nodes and thus higher complexity. The results are visualized in Figure 6.

For the maze tasks, we do not change the size of the maze as it would result in resolution inconsistencies. Instead, we create a dataset of easier samples for training (e.g., if the source and sink points

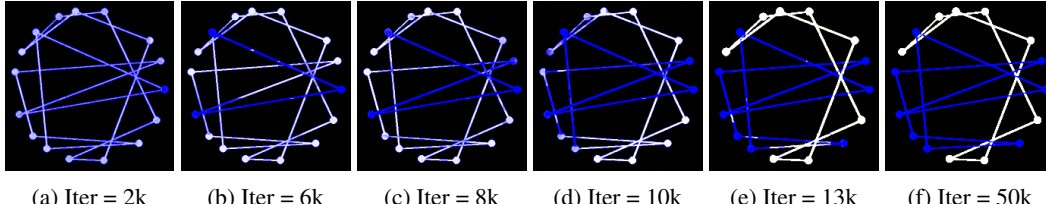

(a) Iter = 2k     (b) Iter = 6k     (c) Iter = 8k     (d) Iter = 10k     (e) Iter = 13k     (f) Iter = 50k

Figure 7: Generated scratchpads for an example at different stages during training. We have increased the contrast of the images for better visualization. It can be seen that the model first learns to color the rightmost node and then it goes one distance further each time during training.

are connected their distance is less than or equal to 30) and use the main task dataset for validation. We explain the OOD training datasets for the maze tasks in more detail in Appendix D. The OOD results for the rectangular maze task are shown in Table 1. It can be seen that the inductive scratchpad model achieves almost perfect accuracy on OOD samples while the single-scratchpad model performs slightly better than random. We present more OOD experiments in Appendix E.

### 4.3 ABLATIONS

The success of the inductive scratchpad model can be attributed to several factors such as increased supervision during training, halting mechanism, and the combination of teacher forcing (TF) and training on the output distribution of the model at training. In this part, we provide experiments that show the effect of these elements. To do this, we consider a multi-frame baseline model which is similar to the single-frame model with the difference that it has multiple heads for predicting multiple scratchpad frames.[5] For the "Inductive w/o halting" baseline we simply set a fixed large number of steps. The OOD performance of different

| Method | ID (%) | OOD (%) |
|---|---|---|
| Inductive | **99.99** | **88.21** |
| Inductive (only TF) | 99.91 | 85.15 |
| Inductive w/o halting | 99.94 | 80.89 |
| Multi-frame | **99.99** | 64.83 |
| Single-frame | 99.95 | 64.79 |

Table 2: Comparison of in-distribution and average out-of-distribution accuracy for multi-frame and single-frame scratchpad variants.

variations on the cycles task is reported in Table 2. It can be seen that removing aspects such as teacher forcing and adaptive (early) halting can each cause minor reductions in the performance of the inductive scratchpad model, whereas, the multi-frame baseline provides no performance gain on OOD samples compared to the single-frame model. More ablations are shown in Appendix E.4.

### 4.4 STAIRCASE LEARNING PHENOMENON

In our experiments with generating a single-frame scratchpad, we observed progressive hierarchical learning over the scratchpad image prediction task. Consider the cycles task as a running example. In the training set, we always color the cycle that passes the rightmost node. In the scratchpad generations, we can observe that the model first learns to color the rightmost node. Then it learns to color the two neighbors that are connected to the initial node. Similarly, at each of the later stages, it learns to color roughly two more nodes (from the two sides). See Figure 7 for a visualization.

These hierarchical learning phenomena have been previously observed and proven in theoretical settings, in particular, in the context of learning sparse Boolean functions where it is known as the staircase behavior (Abbe et al., 2022; 2023a). The staircase phenomenon states that if the target function has some hierarchical structure and is composed of different parts with different difficulties, learning easier parts first can boost learning for the harder parts. To be more precise, assume we have $n$ i.i.d. uniform Boolean variables $x_1, \ldots, x_n \in \{\pm 1\}$. It is well known that the difficulty of learning degree $k \leq n/2$ parity function, e.g., $x_1 x_2 \cdots x_k$ increases as $k$ increases. In particular, if $k = \omega_n(1)$ then learning the parity function is not possible in polynomial time with regular MLPs (and statistical query methods) and learning complexity of degree $k$ parity for constant $k$ scales with $n^k$ (Abbe & Sandon, 2023). However, Abbe et al. (2022) show that functions such as

---

[5]We use a preset number of heads and repeat the last scratchpad frame for heads with a missing frame.

$x_1 + x_1 x_2 + \cdots + x_1 x_2 \cdots x_k$ can be learned in $O(n)$ time. This is because the network can first learn the 'easy' linear component $x_1$. Now, for learning $x_1 x_2$ the model no longer needs to find two variables (which would scale with $\binom{n}{2}$), but it needs to only find $x_2$ since it knows that $x_1$ is in the support and can navigate the search space more efficiently to learn the terms like $x_1 x_i$.

Considering the cycles task in the single-frame scratchpad again, coloring the first three nodes is a low globality function and can be learned easily. Next coloring the next two nodes once the coloring of the first three nodes is learned is a low globality function (similar to $x_1 \cdots x_i$ when $x_1 + x_1 x_2 + \cdots x_1 \cdots x_{i-1}$ is learned). More precisely, define $Y_k$ to be the color of all nodes (and edges) with a distance less than or equal to $k$ from the rightmost node ($2k+1$ nodes in total). $Y_1$ is a local target, moreover, coloring $Y_{k+1}$ correctly once $Y_k$ is learned is of constant globality degree. This staircase structure allows the model to learn $Y_1, Y_2, \ldots$ and finally the complete scratchpad frame sequentially during training as observed in Figure 7. Note that in the example of cycles task the intrinsic staircase structure of the single-frame scratchpad coincided with the multi-frame scratchpad, however, that is not necessarily always the case.

This example shows that the globality-degree does not satisfy the triangle inequality. In other words, we show that for input $X$ and target $Y$ and diverging globality degree (e.g., increasing number of nodes in the cycles task), there exists a single-frame scratchpad $X_1$ such that globality degrees of $X_1$ from $X$ and $Y$ from $X_1, X$ is constant. Thus, a single-frame scratchpad can make both efficient weak and strong learning (through the staircase effect) possible.

## 5 CONCLUSIONS AND FUTURE DIRECTIONS

Here, we summarize the contributions of our paper. (1) We explored the concept of locality and globality in the domain of visual tasks. In particular, we extended the definition of globality degree (Abbe et al., 2024) to vision tasks. Motivated by the latter we introduced four global vision tasks reminiscent of the connectivity task of Minsky & Papert (1969) that are hard to learn for ViT models regardless of their size even for pre-trained models. (2) We further put forward the concept of visual scratchpads, variants of scratchpads methods used in language (Nye et al., 2021) but for vision, that can break the globality degree of tasks by introducing intermediate subtasks of lower globality degree. These subtasks are of the form of visual frames in vision. We show that training models with a single-frame scratchpad supervision can make the introduced datasets learnable as it breaks the globality degree of the initial task. (3) Finally, we introduce the inductive scratchpad model for predicting multi-frame visual scratchpads in a Markovian manner such that each intermediate frame is predicted using only the previous frame. We show that this model can learn the proposed datasets with smaller model sizes while the single-frame scratchpad model fails. Moreover, we show that the inductive scratchpad model has superior OOD performance as it focuses on learning the reasoning steps and can apply them as many times as needed at inference thanks to the adaptive compute time.

**Future work.** With advancements in the field, we expect (global) reasoning to become increasingly essential in the visual domain as well. In particular, we believe models with text and multi-image input and output modalities will enable the integration of reasoning in the visual and the symbolic domains in an interleaved manner. This reasoning capability can be used for new tasks such as solving geometry questions (current systems rely on the symbolic approach), solving visual puzzles (such as the maze introduced in this paper), and understanding high globality images such as maps. It could also improve performance on existing tasks such as autonomous driving. However, suitable datasets for such tasks are not yet readily available. Nevertheless, this paper takes a first step in this direction by introducing datasets that require global reasoning and necessitate a new approach—the use of visual scratchpads. With the increasing prevalence of multi-modal models, it may become feasible to generate visual scratchpad frames through in-context learning and chain-of-thought prompting rather than relying solely on training-time supervision.

In the long term, we believe our work will also influence research on image and video generation. This can be achieved through a more globally consistent modeling of semantics in complex visual scenarios, particularly in dynamic content generation. Such scenarios often involve understanding high-order spatial relationships and global context reasoning across multiple frames. Global reasoning helps maintain coherence and continuity during generation, ensuring that objects, characters, and environments are geometrically consistent and interact naturally within the visual flow.

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

## A  RELATED WORK

We have discussed the most relevant works throughout the main sections of the paper. In this section, we delve deeper into the related literature, examining it from multiple angles.

**Reasoning with Transformers**    In recent years, reasoning capabilities of neural networks and in particular Transformers (Vaswani et al., 2017) have been extensively studied on a variety of topics ranging from completely synthetic symbolic datasets (Zhang et al., 2021b; 2022) to algorithmic tasks (Veličković et al., 2022) and to more natural settings such as mathematical reasoning (Saxton et al., 2019; Lewkowycz et al., 2022). These tasks usually have a combinatorial essence and hence an exponentially large input space which makes memorization-based learning approaches impossible for the Transformers. Another tool for assessing the reasoning abilities of neural networks is to test their OOD generalization performance to see whether they rely on superficial cues that do not work on OOD samples or rather they can compose the rules they have seen during training to generalize to OOD and often more complex examples. As a special case of OOD generalization, it has been observed that length generalization (Zaremba & Sutskever, 2014; Lake & Baroni, 2018; Hupkes et al., 2020), generalizing to longer instances than what seen during the training, is particularly challenging for Transformers even for simple arithmetic tasks such as parity, addition, and multiplication (Anil et al., 2022; Abbe et al., 2023b; Lee et al., 2024). This challenge may be further aggravated in the settings where the input problem or its solution is longer than what the model has seen during training and hence the model has to deal with (mostly) unseen positions where it has been shown the absence or the use of different absolute or relative positional embeddings (Shaw et al., 2018; Dai et al., 2019) result in significant variations on length generalization performance (Kazemnejad et al., 2023). Despite the efforts to understand the reasoning abilities in the symbolic domain, works in the visual domain have focused on shallower types of reasoning emphasizing understanding the semantics of the image. This is despite the fact that vision provides an excellent ground for OOD and length generalization experiments since one can easily depict more challenging examples with the same image resolution which removes the element of using suitable positional embeddings from the picture.

**Visual reasoning.**    Different datasets have been introduced to evaluate various aspects of reasoning in the visual form. For instance, visual question answering (VQA) dataset (Antol et al., 2015) asks questions about an image in natural language. These questions can rely on understanding the semantics in the images and basic reasoning operations such as counting. CLEVR (Johnson et al., 2017) is a diagnostic VQA dataset made up of synthetic objects that removes spurious correlations that models can use in traditional VQA datasets, in addition to disambiguating the types of the errors that the model can make. The reasoning operations considered in CLEVR include counting, comparison, attribute identification, and combinations of those. GQA (Hudson & Manning, 2019) is another VQA dataset with real images focusing on answering compositional questions inspired by CLEVR. VCR (Zellers et al., 2019) is focused on commonsense reasoning, asking deeper questions based on images (e.g., intentions of people and why an event is happening). CLEVRER (Yi et al., 2020) focuses on understanding videos of CLEVR-like objects. In these videos, events such as collisions happen and different descriptive, explanatory, predictive, and counterfactual questions are asked. The CATER dataset (Girdhar & Ramanan, 2020) is focused on temporal reasoning where a video is given to a model and the model's task is to track a particular (potentially occluded) object throughout the video (similar to cups and ball shuffle game). ACRE (Zhang et al., 2021a) is another dataset that aims to assess the performance of vision models in performing causal induction. Winoground dataset (Thrush et al., 2022) also focuses on compositional reasoning. Given two images and two captions with the same set of words, the task is to match them correctly which is shown to be very challenging for vision models. There are also datasets that require reasoning with a physical world model such as the Phyre dataset (Bakhtin et al., 2019). Most of the aforementioned datasets rely on understanding semantics in an image and in contrast to our proposed datasets are easily solvable by humans.

MathVista dataset (Lu et al., 2023) focuses on mathematical reasoning in the visual context. In this case, the questions are a combination of an image and text, however, the reasoning required for answering the question is done in the text domain. Some datasets are inspired by human IQ tests and Raven's progressive matrices (Santoro et al., 2018; Zhang et al., 2019; 2024) that may be more

challenging for humans compared to the classical VQA datasets, however, it is still not clear how one can increase the difficulty and the required number of reasoning steps for these datasets.

More visually similar to us is the Pathfinder (Linsley et al., 2018) which was introduced to show that convolutional neural networks (CNNs) cannot model long-range spatial dependencies well enough. The Pathfinder dataset in the text format was later included in the long-range arena benchmark (Tay et al., 2021) which aims to evaluate Transformers' ability to model long-range token dependencies. We note that our datasets do not necessarily focus on the distance between tokens (or the distance in the image) but rather the globality degree of the task and the number of reasoning steps required to solve the task. To the best of our knowledge, the proposed datasets in this paper are unique in terms of having a scalable globality degree and number of reasoning steps while being challenging for humans as well.

**Scratchpad and chain-of-thought.** Nye et al. (2021) introduced the idea of scratchpads showing that training Transformers to output the intermediate reasoning steps in addition to the final solution can boost their performance on reasoning tasks such as arithmetic, math, and code understating. Further, Wei et al. (2023) show that models can learn step-by-step reasoning by merely seeing a few in-context examples referring to this by chain-of-thought (CoT). Later it was shown that pre-trained language models can generate chains of thoughts only by prompting to do so (Kojima et al., 2023). Abbe et al. (2024) provide theoretical explanations on the effectiveness of scratchpads using the notion of globality concept.[6] They also introduce a variant of the scratchpad method for multi-step reasoning problems that uses a dynamic masking technique to only attend to the input question and last step which causes the model to demonstrate superior length generalization performances.

Moreover, there have been recent efforts to use the visual form of scratchpad and chain-of-thought in multi-modal models. In particular, visual-CoT (Shao et al., 2024) takes an image with a question in the input. During the generation of the output, it first predicts a bounding box in the image that may have important information inside, and then the model focuses on that part of the image to answer the question better. This idea could be useful in cases where the answer can be given using a small part of a high-resolution image (e.g., a text written with a small font in the corner of an image). However, this work does not deal with hard reasoning tasks that require multiple reasoning steps nor produce images as scratchpad/CoT. Concurrent to us is the work of Hu et al. (2024) where they introduce the notion of sketchpad. For a question (consisting of textual and visual components) they use a set of visual operations and tools including drawing lines, drawing bounding boxes with object detection models, and using Python to produce plots to generate a sketch that can potentially facilitate the reasoning process. The main difference between our works is that we focus on visual tasks that have a high globality degree and require multiple reasoning steps to solve, whereas Hu et al. (2024) do not consider visual tasks that require multistep reasoning. As a result, the view in our work is to use visual scratchpads to make the tasks learnable, while in their case is to use tools (e.g., object detection or plot creation using Python) to generate images that can guide the model. As a result, in our case, the models can generate a sequence of frames that correspond to reasoning steps where each image is generated freely by the model. While the sketchpad method can only generate a single sketch in a limited manner by using a set of predefined tools and operations.

**Recurrent architectures.** Several works have introduced a recurrent component into Transformer architectures (Dehghani et al., 2019; Hutchins et al., 2022; Giannou et al., 2023). Notably, Universal Transformers (Dehghani et al., 2019) use shared weights between transformer layers and also uses an adaptive computation time (Graves, 2016) by varying the number of times that the transformer layer is applied. We note that the inductive model proposed in this paper is significantly simpler than the architectures above. This is because in the proposed inductive model, due to the Markovian modeling of scratchpad frames, there is no sort of adaptive compute time involved at train time, and the model is simply supervised to generate the next frame given the current frame without any history (see Appendix C). Further, the halting mechanism is supervised during training.

---

[6]In particular, for the symbolic version of the cycles task studied in Abbe et al. (2024), it is shown experimentally that the learning complexity grows rapidly with the number of nodes ($2n$) increasing.

# B TRAINING DETAILS

We first resize the input (and the scratchpad frames) to $224 \times 224$ resolution. We then use a patch size of $16 \times 16$ to partition the images to 196 patches for all models before giving them to the ViT backbone of the models. The models are evaluated on 10k validation samples.

For training, we use AdamW (Loshchilov & Hutter, 2017) optimizer with weight decay 0.05 and learning rate 0.0003. For the learning rate, we first use a linear warm-up to increase the learning rate from 0 to 0.0003. Afterward, we use a cosine schedule with $3e - 6$ as the end value for the rest of the training. The linear warm-up is applied for $5\%$ of the training time (e.g., 2500 iterations if the total number of iterations is set to $50k$) and the cosine annealing is applied for the rest $95\%$ of the training time.

Each of our experiments has been run on 8 H100 or A100 GPUs and we use a batch size of 1024 for each iteration. The whole project has a approximate total consumption of 160k GPU hours.

## B.1 HYPERPARAMETER TUNING AND SENSITIVITY

Note that we have different settings in our experiments where we vary our methodology, model size, and dataset. This gives rise to a combinatorially large number of experiments that each require their own hyperparameter tuning which is infeasible. Nevertheless, we tried sweeps of learning rate and weight decay for some of our in-distribution settings. We found that our models and methods are relatively robust to learning rates in the range of 0.0001 to 0.0005 and weight decays in the range of 0.01 to 0.1. In particular, we observed that a learning rate of 0.0003 and weight decay of 0.05 work well in all of the tested settings, and therefore we use this combination for all experiments reported in this paper. Similarly, for the batch size we tried batch sizes 1024 and 2048. We observed that batch size 2048 converges with slightly fewer number of iterations, however, longer wall-clock time. Thus, we decided to use batch size 1024 across all of our experiments.

# C MODEL IMPLEMENTATION

We use a ViT backbone for all of our methods. We use four standard sizes for the ViT model: small, base, large, and huge. Different ViT models differ in the number of layers, embedding dimension (hidden size), MLP size, and number of heads, see Table 3 for more details. Note that these model sizes are standard (Dosovitskiy et al., 2020; Touvron et al., 2021), further, we always use 196 patches of size $14 \times 14$ for all model sizes.

Table 3: ViT model sizes and specifications

| Model | Hidden size | Number of layers | Attention heads | MLP size | Parameters |
|-------|-------------|------------------|-----------------|----------|------------|
| ViT-Small | 384 | 12 | 6 | 1536 | $\sim$22M |
| ViT-Base | 768 | 12 | 12 | 3072 | $\sim$86M |
| ViT-Large | 1024 | 24 | 16 | 4096 | $\sim$307M |
| ViT-Huge | 1280 | 32 | 16 | 5120 | $\sim$632M |

Finally, note that currently, the scratchpad frames in our tasks are deterministic. As a result, our image generation models are also deterministic. We expect that for more complicated tasks a random generation model for the scratchpad frame(s) may be more suitable. One can use different solutions in that case. For example, if there is a constant (say 2) number of possible scratchpad frames, the model can try to generate all these possibilities with a bipartite matching loss similar to the DETR work (Carion et al., 2020). Alternatively, one can add a noise variable $z$ for the generation part to add randomness such that the output scratchpad image is conditioned on the input image of the model. Nevertheless, we emphasize that the focus of this work is on the idea of a visual scratchpad and the need for it and modeling choices (e.g., multi-frame inductive model) and not on image generation methods and hence we have used a simple generation method.

### C.1 Training procedure for the inductive scratchpad model

Consider an input image $x = f_0$ with scratchpad frames $f_1, \ldots, f_T$ and label $y$ from the training set. The recurrent module $\mathcal{M}$ can be trained by teacher forcing, i.e., the model can be trained on samples of the type $f_i \to (\hat{f}, \hat{y}, \hat{h}) = (f_{i+1}, y, \mathbb{1}(i+1 = T))$. The issue with this training method is that the recurrent module $\mathcal{M}$ is solely trained on samples from the training distribution. However, during inference where scratchpad frames are not available, the recurrent module $\mathcal{M}$ will use its own generated frames as the input to itself. This discrepancy between the input distribution of the module at training and at test time could deteriorate the model's performance. We initially implemented our model with teacher forcing training described above and observed that the model can learn all the tasks rather well. The issue, however, is that, especially at the beginning of training, the predicted frames are not guaranteed to be close enough to the train distribution to perform well during inference. Hence, we decided to use the following alternative approach. We provide a frame $f_i$ from the training set to the model to get the predicted next frame $\hat{f}_{i+1}$ along with the predicted label and halt variables $\hat{y}_{i+1}, \hat{h}_{i+1}$. We then provide the predicted scratchpad frame to get the next frame $\hat{f}_{i+2}$ along with the next prediction for the label and halt variable $\hat{y}_{i+2}, \hat{h}_{i+2}$. Finally, we compute the loss for all $\hat{f}_{i+1}, \hat{y}_{i+1}, \hat{h}_{i+1}, \hat{f}_{i+2}, \hat{y}_{i+2}, \hat{h}_{i+2}$ and their corresponding values in the training set. Note that we consider $\hat{f}_{i+1}$ an independent input for the model and no gradient is backpropagated through it. As a result, during training the model's input comes from both the training distribution and the distribution of the generated frames of the model itself. We found that this method gives a considerable increase in the training speed of our models and decided to use this method for our experiments.

We note that this problem of discrepancy between training distribution and generation distribution during inference has been previously observed in settings such as text generation in recurrent neural networks (RNNs) and reinforcement learning, for instance in (Bengio et al., 2015) which proposed a scheduled sampling approach as follows: for each token, they sample it either from the train distribution with probability $\epsilon$ or from the model itself with probability $1 - \epsilon$ and use a schedule (e.g., linear or exponential) to reduce $\epsilon$ during training. We note that our setting is simpler as the modeling in our setting is Markovian and each scratchpad frame is only generated based on the previous one and not the whole history in contrast to RNNs. Hence, our simplified approach of having a fixed rate of samples from the training and generation distributions worked well.

We also note that one could use a large predetermined number of steps instead of using a halting mechanism. For this, one needs to supervise the model such that if the final frame is given to the model, the model outputs the same final frame without changes.

## D  Dataset generation

For each task, we generate a dataset with 1M training samples and 10k validation samples. For both validation and training sets half of the samples have 0 and half have 1 as the label meaning that the baseline accuracy for this dataset would be $50\%$. It is important to note that these datasets are generated in a way that minimal spurious correlations are introduced, otherwise, the model might have used those correlations for weak learning and achieving better-than-random accuracies. We explain the generation algorithm for each of the datasets below.

### D.1  Cycles task

The cycles task consists of $2n$ nodes and $2n$ edges such that the $2n$ edges either form a cycle of size $2n$ or two cycles of size $n$. The label for the former is 1 (connected) and for the latter is 0 (disconnected).

For the cycles task, we generate images of size $448 \times 448$. We further choose the nodes randomly on an invisible circle with a radius of 220. Constraining the nodes to be on an invisible circle ensures that no three points are (almost) collinear. In this case, each node on the circle can be specified by its angle $\theta$. We also ensure that every two nodes are at least $\epsilon$ radians apart on the circle. To generate the points, we select $n - 1$ random numbers between 0 and $2\pi - n\epsilon$ and then sort them: $x_1 \leq x_2 \leq \ldots \leq x_{n-1}$. We also select a parameter $\beta$ randomly in $[0, 2\pi]$. Finally, we define the

points to be

$$\theta_1 = \beta, \theta_2 = \beta + x_1 + \epsilon, \theta_3 = \beta + x_2 + 2\epsilon, \ldots, \theta_n = \beta + x_{n-1} + (n-1)\epsilon.$$

One can easily check that $\theta_{i+1} - \theta_i = \epsilon + (x_i - x_{i-1}) \geq \epsilon$ (where we take $x_0 = 0$). Also, $\theta_n = \beta + x_{n-1} + (n-1)\epsilon \leq \beta + (2\pi - \epsilon) = \theta_1 + 2\pi - \epsilon$ showing that each two consecutive points have a minimum distance of $\epsilon$ radians on the circle.

**Scratchpads.** For the multi-frame scratchpad of the cycles task, we first color the rightmost node in blue for the first frame. At each later frame, we color (at most) two more nodes/edges from both sides. In other words, the $k + 1$th frame includes all the nodes/edges with a distance less than or equal to $k$ from the rightmost node colored in blue. Consequently, the last scratchpad frame which is the same as the single-frame scratchpad for this task colors the cycle that passes through the rightmost node in blue (whether the label is 0 or 1). We note that this resembles to what humans would naturally do by following one of the cycles (with a pen for instance).

**OOD samples.** For the OOD experiments, we simply use the cycles tasks with a different number of nodes for out-of-distribution evaluation. We note that currently, we only generate the cycles task datasets with up to 24 nodes. We believe one has to increase the image resolution for a larger number of nodes to still keep the task visually meaningful.

## D.2 STRINGS TASK

The generation process of the strings task is similar to the cycles task. We have $2n$ invisible nodes (called anchor nodes) and these $2n$ nodes are connected with $2n$ 3rd-degree Bézier curves such that we have either two strings (label 0) or a single string (label 1) equiprobably. For this task, we also generate images of size $448 \times 448$ and choose the anchor points on an invisible circle of radius 200 with the same process described for the cycles task.

Next, we explain how Bézier curves are drawn. To specify a $k$th degree Bézier curves between points $A$ and $B$ one needs to first define $k - 1$ control points $C_1, \ldots, C_{k-1}$. To simplify the notation, we define $P_0 = A, P_1 = C_1, \ldots, P_{k-1} = C_{k-1}, P_k = B$. In this case, the Bézier curve is given by

$$\mathbf{B}(t) = \sum_{i=0}^{k} \binom{k}{i}(1-t)^{k-i}t^i P_i = (1-t)^k P_0 + k(1-t)^{k-1}t P_1 + \cdots + k(1-t)t^{k-1}P_{k-1} + t^k P_k$$

for $t \in [0, 1]$. In particular, the cubic Bézier curves between points $A$ and $B$ with control points $C_1, C_2$ is given by

$$\mathbf{B}(t) = (1-t)^3 A + 3(1-t)^2 t C_1 + 3(1-t)t^2 C_2 + t^3 B \quad t \in [0, 1].$$

We need to specify two control points for each Bézier curve. We also want the curve to look continuous to have smooth strings and as a result, we need the first derivative of the curve to be well-defined. Note that the derivative of the cubic Bézier curve above is given by

$$\mathbf{B}(t)' = 3(1-t)^2(C_1 - A) + 6(1-t)t(C_2 - C_1) + 3t^2(B - C_2) \quad t \in [0, 1].$$

More specifically, we need to ensure that the derivatives are the same at the points that two Bézier curves meet, i.e., at $t = 0$ and $t = 1$ where the derivative is equal to $\mathbf{B}(0)' = 3(C_1 - A)$ and $\mathbf{B}(1)' = 3(B - C_2)$ respectively. To define these points, further assume that points $A, A'$ and $B, B'$ are connected with cubic Bézier curves (i.e., we want a continuous curve that passes through $A', A, B, B'$). Also, to disambiguate the control points, we use notation $C_1(X, Y)$ the first control point for the Bézier curve between $X$ and $Y$ (similarly for $C_2$). Given the derivatives computed above, we need to ensure that $C_1(A, B) - A = A - C_2(A', A)$ and $B - C_2(A, B) = C_1(B, B') - B$. To satisfy these conditions we take

$$C_1(A, B) = A + \alpha(B - A'), \ \ C_2(A, B) = B - \alpha(B' - A),$$

for a constant value of $\alpha$. One can easily check that defining the control points with the equation above makes the first derivative of the curve well-defined and the curve continuous. For instance, to check the continuity at $A$ we have

$$C_1(A, B) - A = \alpha(B - A') = A - (A - \alpha(B - A')) = A - C_2(A', A). \tag{1}$$

For our datasets, we use the value $\alpha = 0.25$ as we find it empirically to produce suitable samples.

**Scratchpads.** In order to generate the multi-frame scratchpad for the strings task, we use a similar procedure to what we do for the cycles task. We first color the rightmost anchor node. At each of the later frames, we extend the colored string from both sides by going to the next anchor nodes. Thus, the $k + 1$th frame colors the string that passes through the rightmost anchor node up to the anchor nodes that have distance $k$ from the rightmost starting anchor node. Analogous to the cycles task, the last scratchpad frame (equivalently the single-frame scratchpad) for this task colors the string that passes through the rightmost node in blue. This is also similar to what humans would do by following one of the strings.

**OOD samples.** Similar to the cycles task, we simply use strings task of different sizes (number of anchor points) for the OOD experiments. Also, as one increases the number of anchor points, one has to increase the image resolution to keep the task feasible to solve.

### D.3 MAZE TASKS

First, we explain the logic shared by both the rectangular and circular mazes. Afterward, we discuss the specifics of these two versions. Our mazes always have two parts a source/start cell colored in blue and a sink/end cell colored in red. The source and sink cell are either in one component (label 1) or not (label 0). Both rectangular and circular mazes can be viewed as graphs where each cell is a graph node and two nodes are connected if they are adjacent and there is no wall between them. We first note that each of our maze components is a tree, which ensures that all cells in one component are connected by a unique path. To generate our maze samples, we first generate a maze that is that has a single fully connected component where any two cells are connected by a unique path (the corresponding graph is a tree). Then we select the start and the end cells, and finally, we add a wall to the maze to break the maze into two components. We will next explain each of these parts in more detail.

There are several algorithms for generating a maze with one component. These algorithms differ in their generation speed, the average length of the paths in the maze, and the branching factor of the maze which specifies the average number of branches in the paths of the maze. Considering these factors, we have decided to use Kruskal's algorithm (Kruskal, 1956) for generating the mazes. Kruskal's algorithm starts with a maze where all possible walls are drawn. Then, at each step, the algorithm selects a wall randomly and removes it if the two neighboring cells of this wall were not previously connected. This algorithm is continued until the maze is fully connected. For the start point of the maze, we select one of the cells adjacent to the first wall selected by the algorithm. We then compute the distances of all the cells to the start cell and in particular the maximum distance $d_{\max}$. Then we uniformly choose the target distance in $[d_{\max} - 20, d_{\max}]$, and select the end cell such that its distance from the start cell is equal to the target distance. This approach ensures that the distance between the start and end cells is random and also large enough to make the maze challenging. Finally, we insert a wall in the maze to make two components. If the label is 0 we put this wall in the unique path that connects the start and end cells, otherwise if the label is 1 we put the wall such that the path between the start and end cells remains intact. In addition to that condition, we select the wall that minimizes the difference in the size of the two resulting components (i.e., our goal is to have components of the same size ideally).

**Scratchpads.** To generate the multi-frame scratchpads of the maze datasets we basically simulate a breadth-first search (BFS) from the start cell. We start from the start/source cell and for each scratchpad frame, we color any cell that is at a maximum distance of 10 from the previously colored cells until we reach the end of the maze component or the solution is found. Note that adding cells of distance 1 at each would have resulted in too many frames. What we do for generating the scratchpads is similar to BFS. In particular, if we define $d_{\text{target}}$ equal to the distance to the end cell if they are in the same component and the maximum distance from the start cell otherwise, then the $k$th scratchpad frame colors all cells within distance $\min\{10k, d_{\text{target}}\}$ from the start cell (note that we end the search once the target is reached or the whole component is explored). In this case, also, the single-frame scratchpad is the same as the final scratchpad frame in the multi-frame scratchpad.

**OOD samples.** Generating OOD examples for the maze datasets is more challenging than the cycles and strings datasets since one cannot simply change the maze size as it will cause resolution inconsistencies. Thus, for the maze dataset, we use the same maze size for the training set and OOD

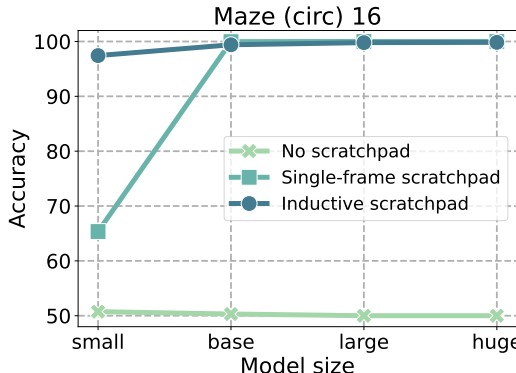

Figure 8: Maze (circular) 16 model size experiments. The model behavior is similar to the maze (rectangular) dataset. Inductive scratchpad is on par with Single-frame for B, L and H but has a significant advantage on S.

samples. Instead, we use *easier* samples for training and use the normal maze task dataset described above for OOD evaluation. To generate easy samples, we choose our target distance between the start and the end cell uniformly from $[10, 30]$ which is significantly smaller than $[d_{\max} - 20, d_{\max}]$ used for the main dataset where $d_{\max}$ was the maximum distance from the start cell (see above). The latter ensures that the number of scratchpad frames required to solve the task when the nodes are connected is less than or equal to 3 during training. Further, instead of trying to split the maze into two components of the same size, we try to add the wall such that the size of the component that includes the start cell is closest to $\frac{30}{d_{\max}}\left(\frac{\text{number of cells}}{2}\right)$. By doing the latter, we make sure that the search space seen during training (size of the component including the start cell) is smaller than the main dataset, and hence samples are easier.

Next, we explain details specific to rectangular and circular mazes.

### D.3.1 RECTANGULAR MAZE SPECIFICS

Rectangular mazes are primarily specific by a number $n$ which indicates the number of rows and columns of the maze resulting in $n^2$ cells. E.g., maze (rect.) 32 has 1024 cells. Also, note that each cell in the rectangular maze has at most 4 neighbors.

### D.3.2 CIRCULAR MAZE SPECIFICS

Circular mazes are organized into a number of concentric rings and are primarily specified by the number of rings. The zeroth circle only includes the center of the maze and is not counted into the number of rings. The first ring contains 6 cells. For each of the next rings the number of cells is kept fixed or is doubled. Also note that the center cell in the circular maze has 6 neighbors and other cells can also have up to 5 neighbors.

## E ADDITIONAL EXPERIMENTS

### E.1 ADDITIONAL MODEL SCALING EXPERIMENTS ON MAZE (CIRCULAR)

In the model scaling experiments conducted on the maze circular dataset in Figure 8, we observe a similar behavior to that seen on maze rectangular. For larger model sizes (Base, Large, and Huge), both the inductive and single-frame scratchpads achieve near-perfect accuracy. However, the inductive scratchpad particularly shines when it comes to smaller models. With the ViT Small model, the inductive approach significantly outperforms the single-frame scratchpad, yielding a performance improvement of more than 30 percentage points. This indicates the effectiveness of the inductive method in handling resource-constrained settings, maintaining superior OOD generalization even when model capacity is limited.

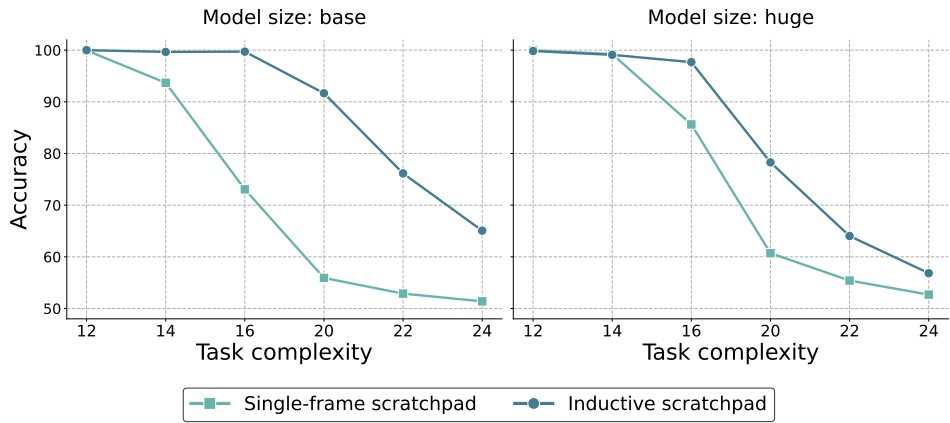

Figure 9: OOD experiment with model size scaling

## E.2 RELATIONSHIP BETWEEN MODEL SIZE AND OOD GENERALIZATION

The plot in Figure 9 presents the OOD generalization performance for models of different sizes (B and H) trained on task complexity 12 and tested on more complex tasks ranging from 14 to 24. Notably, the inductive scratchpad consistently outperforms the single-frame scratchpad across the entire range of task complexities, irrespective of model size. This trend holds true for both the base and huge models, although the performance gap between the two approaches seems to decrease as model size increases. This suggests that the single-frame scratchpad can somewhat benefit from larger models. However, as shown in the main paper, a key advantage of the inductive scratchpad lies in its ability to improve performance by expanding more compute at inference time, enabling smaller models to perform well. We hypothesize that the diminishing gap in performance with the huge model might be attributed to it being more data-hungry. Since the inductive scratchpad sees two steps per iteration for each sample, it may be more prone to memorization, suffering from the additional exposure to images during training.

## E.3 ADDITIONAL OOD EXPERIMENTS ON MAZE (CIRCULAR) AND STRINGS DATASETS

Similar to the experiments presented in the main paper, on the maze circular dataset (see Table 4), both the inductive and single-frame scratchpads achieve near-perfect performance on in-distribution (ID) tasks. However, for out-of-distribution (OOD) tasks, the inductive scratchpad significantly outperforms the single-frame scratchpad, achieving 96.88% accuracy compared to 62.99%. This trend mirrors the results observed on the maze rectangular dataset, where OOD generalization is again much better for the inductive method. For the strings dataset (see Figure 10), the pattern slightly differs. Strings is a more challenging dataset overall, as established in the main paper, which makes OOD generalization particularly difficult. Nonetheless, the inductive scratchpad consistently performs better than the single-frame scratchpad, especially on more complex OOD tasks, with the exception of size 14, which is the simplest OOD task in this setting.

## E.4 ADDITIONAL ABLATIONS FOR THE MULTI-FRAME BASELINE

In the main paper, we discussed the factors contributing to the success of the inductive scratchpad model, including increased supervision during training, the halting mechanism, and the integration of teacher forcing with training on the output distribution. In this section, we present additional experiments to evaluate the impact of the multi-frame supervision. We introduced a multi-frame baseline model, which, while similar to the single-frame model, features multiple heads for predicting several scratchpad frames.

Our previous findings indicated that the multi-frame baseline did not yield any performance gains on OOD samples compared to the single-frame model. However, there is a scenario where the multi-frame approach proves beneficial: it aids convergence for smaller models (Base and Large). As illustrated in Figure 11, the inductive scratchpad converges across all model sizes (Small, Base,

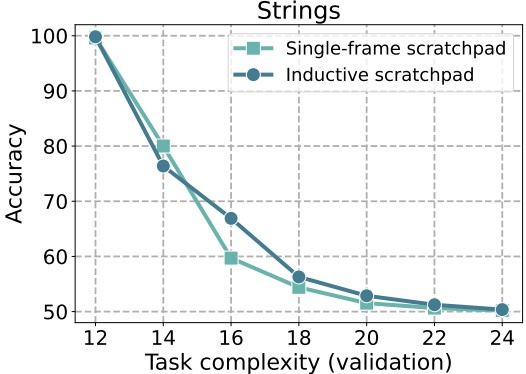

| Method | Accuracy (%) | |
| --- | --- | --- |
| | ID | OOD |
| Single-frame scratchpad | **100.00** | 62.99 |
| Inductive scratchpad | 99.98 | **96.88** |

Table 4: In-distribution (ID) and out-of-distribution (OOD) performance on the maze circular dataset for different methods. The inductive scratchpad outperforms the single-frame scratchpad in the OOD setting.

Figure 10: OOD experiments where the model is trained on strings 12 and tested on more complex strings tasks.

Large, and Huge), while the single-frame scratchpad only converges for the Huge model, indicating a greater computational demand to find the solution. The multi-frame model mitigates this issue by facilitating convergence in Base and Large models, suggesting that while it may still struggle with OOD, as noted in the main paper, it may help model in discovering the solution. This improvement can be attributed to the presence of additional frames, which provide better guidance on the path to reaching the solution.

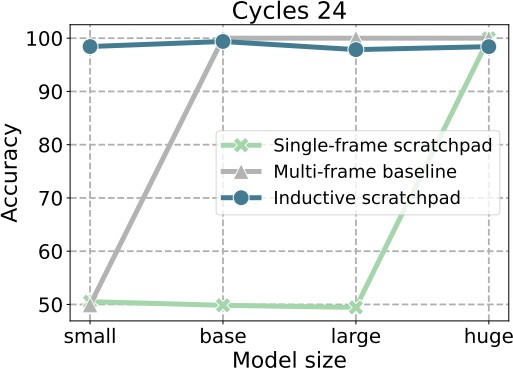

Figure 11: Scaling parameters, Single-frame vs. Multi-frame vs Inductive scratchpad.

### E.5 ADDITIONAL STAIRCASE EXAMPLES

While the main paper presents enhanced visualizations of the staircase phenomenon for clarity, it's important to note that this behavior is also evident in the non-enhanced outputs. As shown in Figure 12, the third row displays the raw model outputs for the cycles 16 task, which exhibit the same progressive learning pattern described earlier. This confirms that the staircase effect is not an artifact of post-processing but a genuine characteristic of the model's learning process.

Moreover, this hierarchical learning phenomenon is not limited to the cycles task. Figure 13 demonstrates that a similar staircase behavior emerges in the more complex maze (rectangular) task. In this case, the model's behavior resembles a spreading "cloud" that progressively discovers contiguous areas of the maze. This is particularly noteworthy because the model is trained only on the final, fully solved maze configuration (shown in the second row of each column), with the first row representing the input maze.

In both the cycles and maze tasks, we observe a consistent pattern of the model first solving easier, more local aspects of the problem before progressively tackling more global structures. This aligns

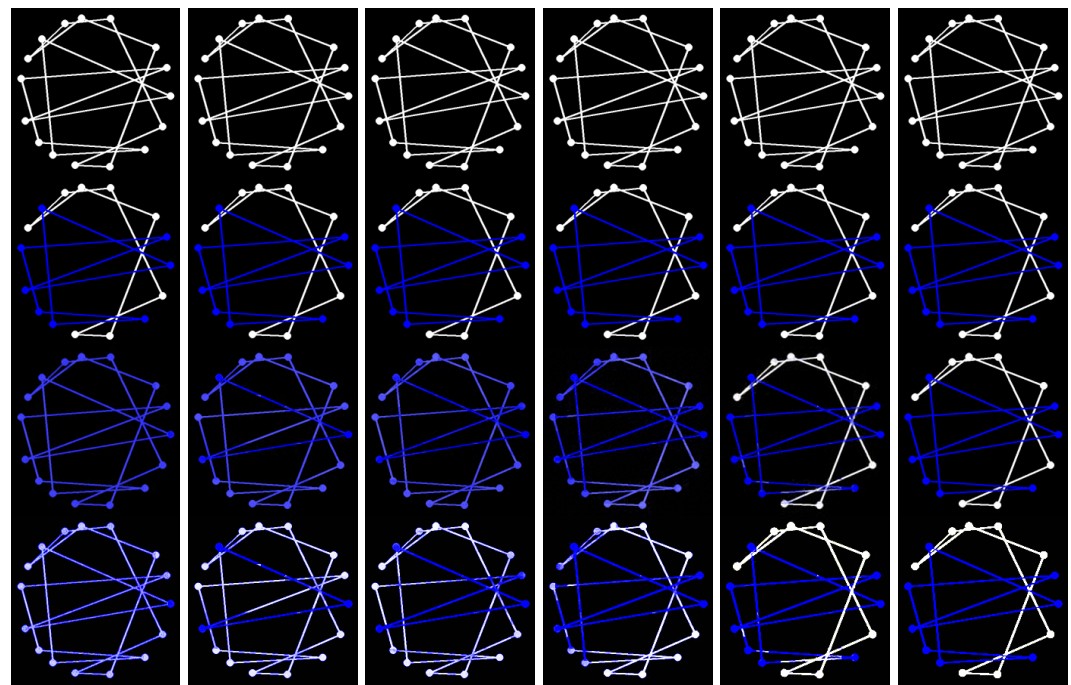

Figure 12: Expanded staircase examples for the cycles 16 task.

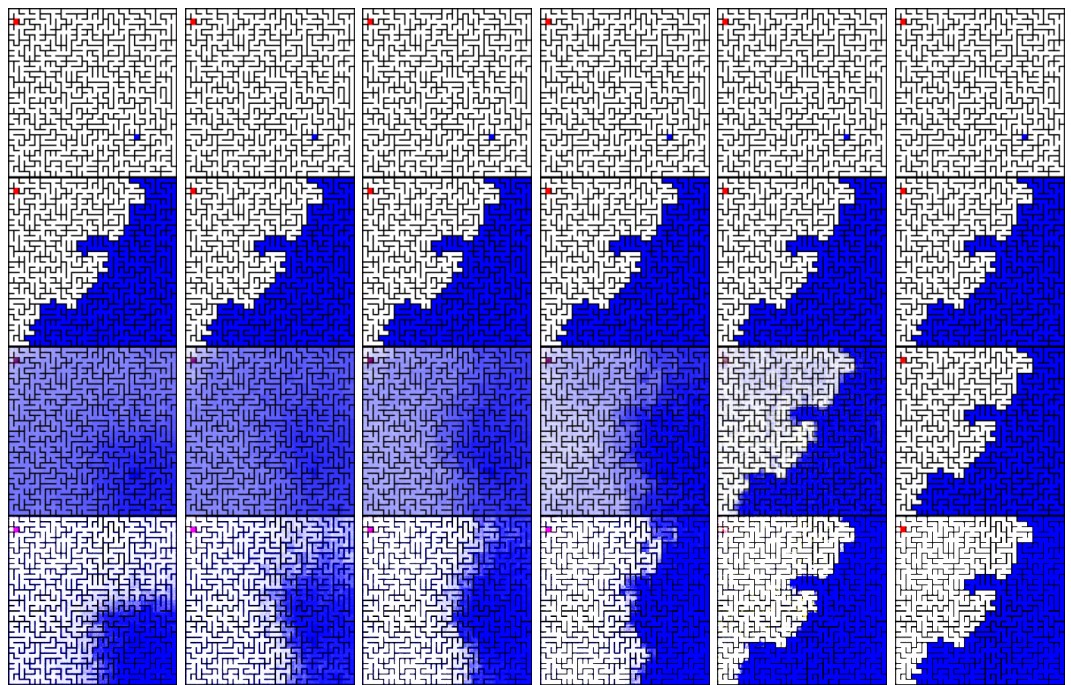

Figure 13: Additional staircase examples for the maze (rectangular) task.

with the theoretical understanding of the staircase effect in learning sparse Boolean functions, but now demonstrated in more diverse and visually complex domains.

Figure 14: The cover of the 2017 edition of *Perceptrons* by Minsky & Papert (1969), which also closely resembles the cover of the 1972 edition. Minsky & Papert (1969) showed that single-layer perceptrons cannot distinguish the two figures based on connectivity due to expressivity issues.

# F  ADDITOINAL FIGURES

## F.1  BOOK COVER OF MINSKY & PAPERT (1969)

Figure 14 shows the cover image of Minsky and Papert's classic book Perceptrons (1969), which explores early theories of neural networks and their limitations.

## F.2  SCRATCHPAD EXAMPLES FOR OTHER TASKS

This section provides example scratchpads for several tasks, demonstrating target frames for the model. The following figures illustrate scratchpads for tasks like connected and disconnected cycles, strings, and solvable and non-solvable mazes. In Figure 15, we show examples of the cycles 20 dataset with connected cycles. In Figure 16 and 17, scratchpads for the strings 12 dataset with disconnected strings are shown. For maze tasks, Figures 19 and 18 display scratchpads for solvable and non-solvable rectangular mazes. Finally, Figures 21 and 20, does the same for maze circular 16.

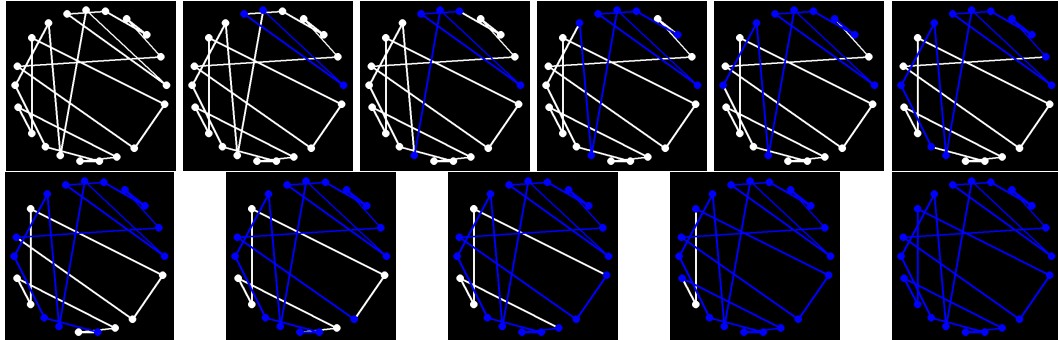

Figure 15: Example of scratchpads for the cycles 20 dataset, connected cycles.

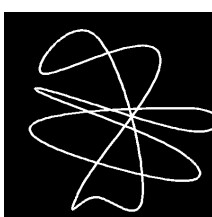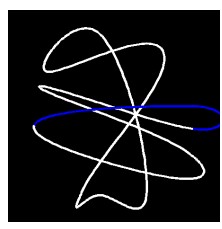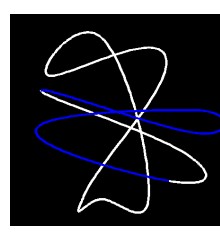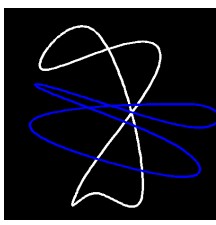

Figure 16: Example of scratchpads for the strings 12 dataset, disconnected strings.

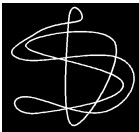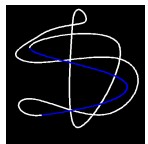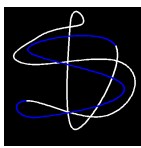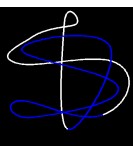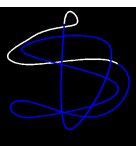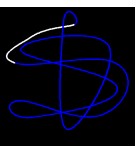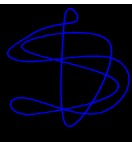

Figure 17: Example of scratchpads for the strings 12 dataset, connected strings.

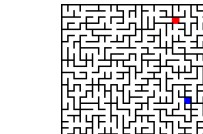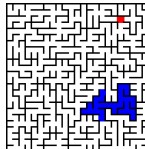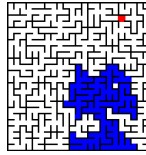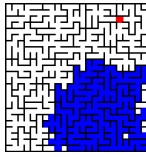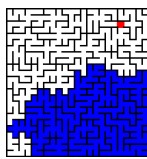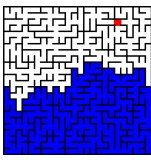

Figure 18: Example of scratchpads for the maze (rectangular) 24 dataset, non-solvable maze.

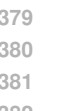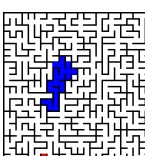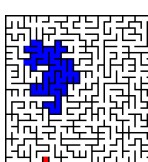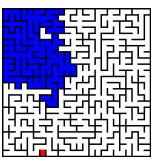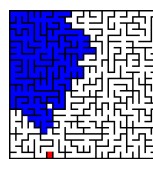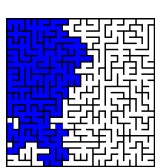

Figure 19: Example of scratchpads for the maze (rectangular) 24 dataset, solvable maze.

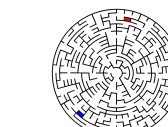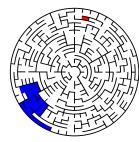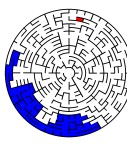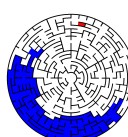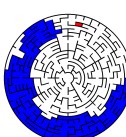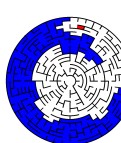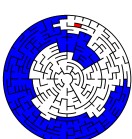

Figure 20: Example of scratchpads for the maze (circular) 16 dataset, non-solvable maze maze.

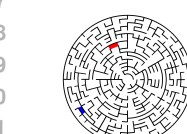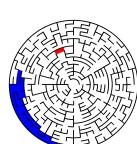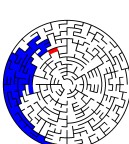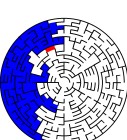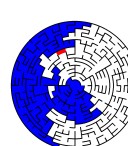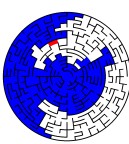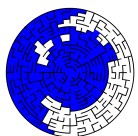

Figure 21: Example of scratchpads for the maze (circular) 16 dataset, solvable maze.

