# OpenReview forum: "Visual Scratchpads: Enabling Global Reasoning in Vision"
_ICLR.cc/2025/Conference — Submitted to ICLR 2025_

### Official Review · Reviewer_MeyH · 2024-10-21

**Soundness:** 2
**Presentation:** 2
**Contribution:** 2
**Rating:** 3
**Confidence:** 4

**Summary:**

The authors propose “global vision tasks”, which studies problems in which reasoning about the whole image is important. They develop a set of datasets involving binary prediction of graph connectivity, string connectivity, and maze connectivity. The paper introduces a method based on a ViT backbone that uses different types of intermediate supervision in the form of a scratchpad (single-frame scratchpad and recurrent multi-frame scratchpad).

**Strengths:**

I appreciate that the authors tackle a complex challenge of reasoning about graphs, strings, and mazes; I think it is an exciting direction that current VLMs struggle at, which we need a general solution for. I also like that the authors evaluate out-of-distribution generalization capabilities.

**Weaknesses:**

1. I have some fundamental disagreements with this categorization of “global visual tasks”. How much “global” vs “local” reasoning is required in a task is on a continuous scale. It depends on not only the task, but the specific instantiation of images and query. The authors point out that “a single patch containing cat whiskers significantly increases the likelihood that the model will classify the image as a cat”. It does, but we will need the full image to understand if the whisker belongs on the cat, or if the whisker instead belongs on a walrus, or if the whisker is instead in a photo on the wall where actually, a person is next to. Similarly, for tasks that the paper is exploring, by seeing that the entry/exit has an immediate connecting path instead of ending in a dead end also increases the likelihood that the maze has connectivity between the two points. How does one determine which tasks are “global” vs. not? The definition is ambiguous, and seems by the authors’ definition to depend on whether a few patches are informative enough to yield high probability predictions. What is considered high probability? On which model, trained on which tasks? How big and how many are these patches? Unfortunately, this task definition is not clear enough, though I appreciate the authors’ attempt to define a more challenging set of reasoning tasks.
2. The method proposed improves upon the baseline because it has more supervision. The paper states that having scratchpads improves performance compared to having no scratchpads, but we can not disentangle whether having a scratchpad in the model forward pass is more important, or whether the supervision is important, as each method is given different levels of supervision. The no scratchpad baseline is given only the final binary answer as supervision. The proposed single scratchpad model is given supervision on what the scratchpad should be. The proposed multi-scratchpad model is, I believe, given 50% of perfect frames for reasoning steps. (a) This does not convince me that scratchpads are helpful, but that intermediate supervision is helpful. (b) What happens when these intermediate frames are not available? Can you generalize to other complex reasoning tasks?
3. The tasks proposed are all quite similar (binary classification, connectivity tasks in graphs, strings, and mazes). I would like to see performance on other vision challenges, even if it’s on synthetic data as well, like ARC.
4. No baselines other than the base ViT are explored on this task. I would expect other visual reasoning prior works to be able to be evaluated.

**Questions:**

See the above four points. I don’t believe this paper is ready for publication in its current form. Though I think the tasks proposed and the broad idea of visual scratchpads is interesting, I don’t think the tasks are well defined, nor the experiments sound enough at this current stage. I would encourage more experimentation on what it means to endow a model with a scratchpad, as compared to additional supervision.

---

> ### Author Response · Authors · 2024-11-21
>
> **Meaning of globality and globality degree.** We first need to clarify the meaning of globality. *The definition of globality is independent of the model.* Globality is the minimum number of tokens to get non-trivial information about the output. This is stronger than just saying that the function requires a global view of the input, it requires a global view to even get a non-trivial partial information.  In the ImageNet example, seeing whiskers in one patch increases the probability for both cat and walrus classes, however, one cannot be certain yet. Hence, the whiskers patch already provides weak learning (i.e., significant information on the label) but not strong learning. *This is enough for the task to be of low globality.* Our tasks are specifically designed to be hard (e.g., spurious correlations are removed). For instance, it is highly unlikely that one can get some signal about the maze by just looking at one patch (if there were some shortcuts like that, models would have been to weakly learn the task without scratchpads).
>
> Consider a task learned by Transformers such that we have input tokens $X\_1, \\ldots, X\_n$ and the output is $Y$ (e.g. the label). In this case, the globality of the task (given by the distribution of input and output) is well-defined as the minimum number of tokens $k$ such that there are $k$ tokens, $X\_{i\_1}, \\ldots, X\_{i\_k}$, that along with the histogram of the tokens ($\\hat P\_X$) provide non-trivial information on the output. I.e., $$I(X\_{i\_1}, \\ldots, X\_{i\_k}, \\hat P\_X; Y) \= n^{-O\_n(1)}.$$
> *We clarify that the globality degree is indeed defined for the distribution of (X, Y) and not for specific samples (similar to how learning is defined for the distribution of inputs and outputs; while each example may be predicted correctly or not).* We note that it has been conjectured and proven in certain cases in \[Abbe et al. 2024\] that if globality is not constant (in $n$) then the task is not learnable in $poly(n)$ time by a $poly(n)$-size Transformer.
>
> Now, in the visual domain, the number of tokens (number of patches) is often constant in contrast to text. (Although one could for instance consider mazes of increasing size and resolution). When the number of patches, $n$, is constant, one cannot define terms asymptotics $poly(n), n^{-O\_n(1)}$ properly. Nevertheless, from the asymptotic results in the general case, we know that as tasks get more global, larger models and more training iterations are required to learn them. We believe in practice, one can easily compare the globality of tasks by just inspecting them, or by running some simple experiments to see how masking patches of the images affects learning, as we compared ImageNet to our cycles task.
> This is why in Definition 1, we add a threshold $\\alpha$ to our definition. For a fixed $\\alpha$, one can always compare the globality degrees of two tasks.
>
> Are there parts of the definitions of globality and tasks that are still unclear?
>
> **Role of supervision.** So there are two comparisons here.
> (1) Comparing the no scratchpad model to the model with a single-frame scratchpad, the models are almost identical and their expressive power is the same. The improvements are indeed coming from having more supervision as the task is broken into subtasks that are easier (have lower globalities). The comparison methodology here is similar to the original scratchpad paper \[Nye et al. 2021\].
> (2) Comparing the single-frame model with the inductive model; here it is not only the amount of supervision that matters but the way the supervision is fed in, i.e., the inductive format of the scratchpad. *We have already explored this in Appendix E.4.* Here we consider a simple model that predicts all frames in parallel, in contrast to our inductive model. We show that this model has better in-distribution performance compared to the single-frame model which shows the role of more granular supervision. Nevertheless, we show that the performance of the inductive scratchpad on both in-distribution and especially OOD samples is significantly higher, which shows the importance of making the supervision in a more compositional manner.
>
> For some tasks, the intermediate frames may not be available. We note that this is not a new challenge specific to our work and it applies to all scratchpad methodologies \[Nye et al. 2021\]. For example, when people supervise math questions with both detailed solution steps and the final answer, one can wonder what happens when the solution steps are not available. In general, one should try to supervise the scratchpad at least for a portion of the data. Note that scratchpad data often exists naturally, e.g., if one takes videos from the web.

---

> > ### Author Response · Authors · 2024-11-21
> >
> > **Other datasets.** We note that to our knowledge, the proposed datasets are the first ones where the source of complexity is coming from the high globality of the tasks. Our tasks are not weakly learnable for rather large models and training sets. This is in contrast to the vision benchmarks we are aware of. For instance, in the ARC challenge, the main sources of difficulty are the limited size of the training set and the fact that the test set is harder than the train set (OOD generalization). However, there is a global (or multi-step) reasoning process, and probably the task would be learnable if the training set was larger and there was no mismatch between train and test distribution. (On a side note, most works that get good scores on ARC do not regard it as a visual task and use text tokens.)
> > We believe our methodology would be beneficial for some other challenging reasoning tasks, e.g., solving geometry questions, in which the solution can be developed using several scratchpad frames. However, such datasets are not readily available at the moment.
> >
> > **Other visual reasoning baselines.** We picked ViT models of various sizes as our baselines since they are the backbone of modern vision architectures. From the theoretical results of \[Abbe et al. 24\], we expect our proposed tasks to be challenging for any architecture with invariance properties. We are not aware of any models which are different enough from ViTs and could be for some reason suitable for our proposed tasks. Does the review have any particular baseline in mind?
> >
> > Please let us know if any part is still unclear. We believe we have answered most of your concerns and we hope you consider increasing your score accordingly.

---

> > > ### Author Response · Authors · 2024-12-02
> > > **Please check our response**
> > >
> > > Dear Reviewer MeyH,
> > >
> > > We hope that our responses have adequately addressed your concerns and clarified any misunderstandings. As the discussion phase is coming to an end, we would appreciate any further feedback or questions you might have.
> > > We hope you might reconsider your rating in light of our clarifications.

---

> > > > ### Comment · Reviewer_MeyH · 2024-12-02
> > > > **Response to Authors**
> > > >
> > > > Thanks to the authors for providing clarifications to my response as well as other reviewers. I am keeping my score as is, (1) I think the evaluation tasks are highly limited (to cycle-based tasks, as pointed out by Reviewer Nq7d), and (2) the novelty of the method itself is limited (as also pointed out by Reviewer 83P1), and (3) the role of supervision here is understated (especially between the no scratchpad model and the model with a single-frame scratchpad). I thank the authors for pointing to ablations between a simple model that predicts all frames in parallel, in contrast to their inductive model. However, I think there are many other ways of leveraging this dense supervision for models as baselines, and in addition, requirement for such supervision leads to highly limited settings where the method could possibly be applicable. I believe the paper could be strengthened in a future iteration with more robust evaluation.

---

> > > > > ### Author Response · Authors · 2024-12-03
> > > > >
> > > > > Dear Reviewer MeyH,
> > > > >
> > > > > Thank you for engaging in the discussion. We hope to continue this dialogue before the discussion phase ends. We are glad the concept and definition of globality have been clarified.
> > > > >
> > > > > (1) **Tasks:** It is true that our datasets share the theme of connectivity. This was by design, as we aimed to focus on tasks where globality is in its most extreme form and where we could control task complexity and data generation process. This allowed us to create image classification tasks that are not even weakly learnable. This is itself novel and in contrast to prior visual benchmarks, as far as we know. Although the range of datasets is limited, they represent a first step that effectively demonstrates the central message of our work --- the necessity of scratchpads in the visual domain. We believe the insights gained from this work are valuable and worthwhile to share with the community.
> > > > >
> > > > > (2) **Novelty:** You have mentioned that the work's novelty is limited. Would you please elaborate? We have clearly outlined the differences with prior work suggested by Reviewer 83P1, where basically the only shared aspect is the idea of modeling sequences of images. As detailed in our response to Reviewer 83P1, there are important differences in our modeling which leads to superior performance. Moreover, the central message that scratchpads are essential in the visual domain, akin to text, is entirely new. We would be happy to further discuss the novelties of our work if you could specify which parts seem unoriginal and have appeared in prior work.
> > > > >
> > > > > (3) **Supervision:** Could you elaborate on your comment that “the role of supervision is understated, especially between the no-scratchpad model and the model with a single-frame scratchpad”? The architectures and expressivity of the no-scratchpad and single-frame scratchpad models are nearly identical, so the performance gains are indeed attributable to the additional supervision. Our experiments, alongside the theoretical results in [Abbe et al., 2024], demonstrate that even weak learning of high globality tasks is impossible without such extra (low globality) supervision. We believe our approach of teaching the model to predict this supervision is a natural extension of scratchpad concepts from text to vision. We do not think there is any way of crafting meaningful baselines with equalized supervision; similar analyses were also absent in the original scratchpad paper [Nye et al., 2021]. Finally, frame-by-frame supervision is naturally available in video data, which makes this approach practical once such data is collected. We will include this discussion in our revision.

---

### Official Review · Reviewer_83P1 · 2024-11-01

**Soundness:** 3
**Presentation:** 3
**Contribution:** 2
**Rating:** 6
**Confidence:** 4

**Summary:**

The paper argues that most existing vision tasks can be solved by considering only local information in an image. Inspired by this, the paper introduces two kinds of synthetic tasks that require global information: a maze task, where there goal is to find a path through a maze; and a graph connectivity task, where the goal is to determine if a graph is connected or not. The paper also introduces a variant for each of these tasks: a circular maze, and a smoothed depiction of the graph. The paper then demonstrates that the tasks can be solved by training an image generation model on sequences of images that visually represent the incremental generation of a solution (such as the sequence of images showing an incremental breadth-first floodfill for the maze task).

**Strengths:**

The paper is quite well-written and easy to follow. It makes the important argument that considerations regarding "System-2"-style reasoning should not only apply to purely textual tasks but also to other domains, such as vision. The graph and maze tasks are simple visual tasks for initial investigations in this direction (with some caveats below).

**Weaknesses:**

The proposed method, "visual scratchpad", seems to be largely identical to methods like the one proposed in (Yang et al. 2024) or (Bai et al. 2023), applied to a much simpler, synthetic task domain. It seems also related to Procedure Cloning (Yang et al. 2022) and the method described in (Lehnert et al. 2024), although it uses pixels instead of tokens to keep track of previously visited states (then again, that seems to make it a special case of the above-mentioned methods based on video generation).

Since these image-generation methods are able to solve surprisingly difficult tasks just by predicting images, would we not expect them to be able to behavior-clone a floodfill-type solution to maze tasks when training on a large training set? I may be missing something here and will be happy to revise my score if so.

The introduced datasets are somewhat simplistic and reminiscent of existing (but much more comprehensive) visual reasoning datasets, like (Cherian et al. 2023).

Yang et al. 2024: "Video as the New Language for Real-World Decision Making"

Bai et al. 2023: "Sequential Modeling Enables Scalable Learning for Large Vision Models"

Yang et al. 2022: "Chain of Thought Imitation with Procedure Cloning"

Lehnert et al. 2024 "Beyond A*: Better Planning with Transformers via Search Dynamics Bootstrapping"

Cherian et al. 2023: "Are Deep Neural Networks SMARTer than Second Graders"

**Questions:**

One could argue that a natural OOD setting to consider for the tasks (especially the maze task) is variable-size. The authors argue that this would result in "resolution inconsistencies". While it would indeed require careful considerations on the vision architecture to make it resolution independent, I believe it could make the tasks much more interesting and bring them in line with the textual arguments the paper makes on global-vs-local tasks and also the existing literature on length generalization, especially in light of the discussion on Globality Degree in the paper.

I find it a bit surprising that pre-training is required (Figure 4).

Besides the two image generation-based models mentioned above, how does the proposed method relate to "The Predictron: End-To-End Learning and Planning" (Silver et al. 2017)?

The term "Visual Scratchpad" could be slightly misleading, as it seems to suggest an approach that equips a language model with a modifiable visual buffer to support reasoning. An approach like that (with the same name) is discussed in "Can Visual Scratchpads With Diagrammatic Abstractions Augment LLM Reasoning?", Hsu et al. 2023

---

> ### Author Response · Authors · 2024-11-21
>
> We thank the reviewer for bringing these works to our attention.
>
> **Yang et al. (2024) and Bai et al. (2023)** consider video as a unified interface for diverse tasks, leveraging frame-by-frame generation for broader real-world decision-making tasks like robotics and self-driving cars​. They also show some initial reasoning capabilities of such models. There are indeed similarities on the modeling front, as the frames of the scratchpad in our case can be viewed as frames of a video too.
> Our contributions are however different from these prior works.
>
> 1. We propose (high-globality) image classification tasks that are impossible to weakly learn for the current models (even by large models trained on extensive data) and expose fundamental limitations of vision models.
> 2. We show that these tasks are only learnable if a visual scratchpad is used. On the scratchpad front, we study different models, single-frame, multi-frame baseline, the inductive model (with different variations), and we show how different modeling assumptions affect in- and out-of-distribution generalization performances. The whole study on the OOD part is new as well.
> 3. We extend theoretical concepts like globality degree and provide experimental evidence that directly connects task globality to model performance.
>
> In short, the **necessity of visual scratchpads** for certain image classification tasks and the reason for this requirement has been absent from the prior work. The idea that we need scratchpads in the vision domain has not been proposed in these works.
>
> **The dataset of Cherian et al.** also considers vision (alongside language) tasks that require reasoning, including some graph tasks. However, we note that, with having globality in mind, our tasks are designed in a way that there is no shortcut or spurious correlation that the models can use. Thus, the globality of our tasks is high. For our tasks, large models trained with extensive data cannot even achieve weak learning. However, in the work of Cherian et al. we can see that models achieve modest performance in different task categories. Further, our tasks all have adjustable difficulties. Also, our tasks are simple image classification while Cherian et al. is more similar to a VQA dataset.
>
> **OOD on bigger mazes.** We agree that the idea of handling bigger mazes is interesting. There are two possible directions of approaching this: (1) One idea is to keep the number of patches constant and resize larger mazes (higher resolution images). In this case, smaller mazes will be shown larger (zoomed-in) and vice versa. Since we are training these models from scratch, it is highly unlikely that the model achieves good performance on differently sized mazes. However, if one uses pre-trained models and also tries OODs like (train sizes: 10, 12, 14, 16 and test: 18, 20\) one could potentially end up with robust enough models capable of generalizing. (2) Not resizing and instead increasing the number of tokens (similar to Transformers for text). Disregarding the fact that this would increase the FLOPs of the model, making comparisons unreliable, one also has to handle the length generalization similar to text by using suitable positional embeddings, etc. We believe both of these directions are out of the scope of this work. In this work, we are showing that the inductive method is capable of generalizing to harder problems of a given size. Our approach captures what would happen with real-world datasets. Say, if we had a geometry dataset, the resolution and image size would have been almost similar throughout the dataset, however, difficulties would vary. This is what we have done for OOD experiments as well.
>
> Regardless of these design choices, we do not believe these analyses to be relevant for the paper, as we want to isolate algorithmic reasoning, and therefore we design a setting where the atomic building blocks do not change much as that could introduce additional distribution shifts that could act as confounding factors.
>
> **Why pretraining is helpful in Figure 4?** Pretraining is essentially providing a better initialization compared to the model from scratch, which may help optimization. When initialized randomly, the gradient of the neural network is essentially uncorrelated with the high globality target, which makes learning hard. However this may change when we use a pre-trained model and the weights are not random (i.e., pre-training provides a better initialization.) Moreover, much as the prior we add to the model by using BFS algorithms, pre-training provides human priors from the internet-sourced data the model was trained on. For instance, the fact that adjacent patches are likely more related than patches that are far away from each other, or the concept of a “graph” or a “maze”.

---

> > ### Author Response · Authors · 2024-11-21
> >
> > **How is this work related to “The Predictron” by Silver et al. 2017?** Although there are some similarities between our paper and [Silver et al. 2017] (e.g. benchmarks choice and Markovian reward), the premise and purpose of the two papers is quite distinct. [Silver et al.] focuses on learning to plan by using reinforcement learning. While this is an interesting problem, and would possibly help solving our maze tasks as well, that is not the contribution of our paper. We instead focus on understanding what makes tasks hard to learn and what type of supervision mitigates that. Moreover, introducing more complicated methods, like RL, in our analysis would not help understanding where complexity stems from, rather, it would make the system harder to probe and analyze. We hope the reviewer agrees with us on this. Further, note that our datasets are designed specifically to be free of any spurious correlations (or shortcuts) which would allow a model to weakly learn the task. This is not clear for the maze dataset generation used by Silver et al. Finally, although the paper is well-grounded, it does not propose any theory regarding globality nor related experiments. We are happy to include this analysis in the related work.
> >
> > **The term visual scratchpad can be misleading.** Thanks for the feedback. We originally decided on this name as we believed it was the same concept of scratchpad, applied in the visual domain and we wanted to avoid coming up with a new name for a concept that already exists. However, we see that this could cause confusion. We are open to suggestions from the reviewer.
> >
> > Please let us know if any part is still unclear. We believe we have answered most of your concerns and we hope you consider increasing your score accordingly.

---

> > ### Comment · Reviewer_83P1 · 2024-11-26
> >
> > Why would Yang et al. (2024) or Bai et al. (2023) not work on the proposed tasks, given that those methods are identical to the method proposed here? They're not called "Visual Scratchpad" and instead "Visual Sentences" or "Intermediate Frames", but they are otherwise the same.
> >
> > And they were shown to work well on tasks with a high degree of locality and were even argued to be used on mazes in Yang et al. (2024).
> >
> > What is it about those methods that you would except them to fail in your setting? Is it the choice of hyperparameters? The choice of vision encoder (ViT/CNN/...)? Something else?

---

> > > ### Author Response · Authors · 2024-11-30
> > >
> > > We thank the reviewer for engaging in this discussion. While there are conceptual similarities between our method and those proposed by \[Yang et al. 2024\] and \[Bai et al. 2023\], our objectives differ significantly. Moreover, our approach to frame sequence modeling introduces key advantages that are essential for effectively learning the visual scratchpads. We explain these differences below.
> > >
> > > **Difference in objective.** The main objective of our work is not to provide a new framework for modeling sequences of images. Our paper is a study of the globality concept in the visual domain where we show the necessity of the scratchpad idea as a way to counteract globality. I.e., for certain image classification tasks, we need to break the task into subtasks and make the model learn these subtasks using a visual format of the scratchpad. On the other hand, the purpose of \[Bai et al. 2023\] and \[Yang et al. 2024\] is more general, they want to provide a general interface for modeling sequences of frames (and in particular videos), to make a unified interface for different visual tasks, pre-training more useful, and ideas such as in-context learning possible.
> > >
> > > To draw an analogy, the original scratchpad work by Nye et al. (2021) was not about introducing a new sequence modeling paradigm for text but about demonstrating how intermediate reasoning steps improve task performance. Our work similarly brings the scratchpad concept into the visual domain.
> > >
> > > **Difference in methodology.** We assume a Markovian structure between scratchpad frames, where each frame is derived only from the preceding frame. However, neither of \[Bai et al. 2023\] nor \[Yang et al. 2024\] use such an assumption (which is reasonable as their goal is to model arbitrary sequences of frames). The Markovian assumption in our work has important advantages:
> > >
> > > 1. **Better OOD performance:** In the ablation studies of our paper, Section 4.3 and Appendix E.4, we have compared the Markovian modeling of scratchpad frames (the inductive scratchpad proposed in our work) and a baseline without the Markovian assumption. We have shown that the Markovian modeling makes models **capable of OOD/length generalization** while the baseline method without the Markovian assumption cannot generalize to OOD samples.
> > > 2. **Smaller model sizes:** We have further compared the inductive scratchpad (applying the Markovian assumption) with the baseline multi-frame model (scratchpad without the Markovian assumption) in Appendix E.4, and we have shown that the Markovian model allows us to learn the task with smaller models.
> > > 3. **Scalability:** Thanks to the Markovian assumption, our method supports an unbounded number of scratchpad frames. In contrast, in the non-Markovian approaches modeling multiple frames becomes increasingly challenging. In particular, \[Bai et al. 2023\] is limited to 16 frames due to context-size limitations.
> > > 4. **Multi-frame generation:** Unless the task is inherently sequential (e.g., predicting a video), both \[Bai et al. 2023\] and \[Yang et al. 2024\] usually generate only a single image to solve visual tasks (mostly in the in-context learning setting). This is in contrast with our work where we advocate for multi-frame scratchpads for image classification tasks. We also address and mitigate the train-test discrepancy that arises in multi-frame generation via an alternating training process that improves the performance, as shown in Section 4.3.
> > >
> > > In summary, while the suggested papers model sequences of frames and may work in-distribution, they would not generalize to OOD samples and suffer from scalability and efficiency limitations. (Note that their goal is not about breaking visual tasks into subtasks using a scratchpad in the first place.) OOD generalization and managing to have smaller models are key vignettes of a better reasoning model, which one would not achieve with a naive scratchpad. These distinctions make our work uniquely suited for tasks requiring visual reasoning through intermediate subtasks, beyond the scope of prior methods.
> > >
> > > We will revise the related work section to include this discussion and further clarify these distinctions.

---

> > > > ### Comment · Reviewer_83P1 · 2024-12-02
> > > >
> > > > I'm still on the fence regarding the novelty of the method itself (as opposed to the theoretical argument of extending Abbe et al. to the visual domain), especially in the fixed resolution setting. But I believe that adding the qualifications in the difference of objective and methodology will help make the paper stronger and I will raise my rating to 6.

---

### Official Review · Reviewer_Nq7d · 2024-11-02

**Soundness:** 4
**Presentation:** 3
**Contribution:** 2
**Rating:** 5
**Confidence:** 4

**Summary:**

The authors show that tasks that have a high "globality degree" are hard for transformers to learn. Specifically, tasks globality degree is the proportion of patches required to get non-trivial performance on the task. In the language of RL/imitaiton learning, the idea is that tasks with a high globality degree are harder to learn when using sparse rewards, and that process supervision/behavior cloning can help when each action is sufficiently "local".

The paper builds on the results of Abbe et al, and shows results on images. All image tasks are some form of cycle detection and the process supervision is some form of BFS depicted in an image (e.g. a drawing of a maze, a drawing of the nodes/edges of a graph, etc).

------ Previous ----
The authors suggest that an important class of reasoning problems rely on breaking down a complex ("global") problem into a sequence of simpler steps that are more "local". They introduce a measure of "globality" based on what proportion of patches are required to solve the task.

Then they show some experiments on 4 visual depictions of problems where BFS is the solution. They train recurrent transformers to solve this problem, and show that using behavior cloning to approximate the BFS step results in better generalization to new sequence lengths.

They show that in some cases, networks can also learn with less decomposition and chain-of-thought training. Recurrent transformer model trained on all intermediate steps of visual BFS (inductive scratchpad) learns to generalize to samples of different lengths than a model that is trained to go 1→penultimate step, and then penultimate → ultimate step (single-step scratchpad).

They show that the single-step model is able to learn some tasks, provided the model is sufficiently “large” — vit-S up to VIT-H. In this case the model needs to learn to do O(24) steps, and only models with more layers (VIT-B has 12 and VIT-H has 30) end up learning the solution. While for inductive versions, all versions learn the solution. This is probably related to the idea of “effective depth” — e.g. Feedback networks CVPR 17 http://feedbacknet.stanford.edu/.

**Strengths:**

The experiments are rigorous and, to summarize the rebuttal discussion, I agree they relate to globality degree.

-- Previous --

**Originality**
The authors define globality degree in a pretty intuitive way: what proportion of images patches are required to predict the label. They show that for tasks that seem to have a high globality degree (e.g. counting connected components), networks trained with heavy input masking cannot solve the task.

**Significance**
Unclear to me what the significance of globality vs the scratchpad vs making this work for visual settings. It would help if this were connected to vision settings where other papers have studies, rather than a new benchmark of visual representations of BFS

**Clarity**
Generally the text was written clearly and easy to read. Figures were well presented and grounded in the text.

**Quality**
The authors clearly invested significant care and effort in preparing the experiments and writing the manuscripts. The figures are quite visually appealing, too.

**Weaknesses:**

**Experiments:**

-- Final--
The experiments in the paper are on visual tasks that are not very representative of computer vision tasks.

I understand that the task is defined is on tensors of shape B x W x H x C, the inputs are continuous, and the architecture is a ViT. But the images are unlike natural images. Other visual tasks might be a better fit -- e.g. tasks that use imitation learning from pixels. In fact, the method is pretty similar to imitation learning.


-- Previous --
My main concern is that the experiments don't really support the main story of the paper. They seem related at a glance, but after thinking about it some more I don't think they actually are.

What does the globality/locality degree have to do with the experiments? Do transformers learn local steps local steps preferentially to global ones? BFS is local, but they don’t have any experiments or theoretical analysis that it is an important property. Besides, since transformer attention is global anyway, I would expect not.

In fact, the experiments seem to instead show that when the number of attention layers is fewer than the number of "reasoning" steps required for the underlying graph algorithm, the model fails.
* This would explain why larger models (more layers) can learn to do BFS for fixed-size problems even in the single-step case. Specifically: if the # layers > # steps, the model can learn a solution, which would explain why ViT-S and VIT-B fail (they have 12 layers and there are O(24) reasoning steps required).
* It would also explain why single-step models don't learn to generalize as well, even for the deeper models: because single-step models have a fixed number of layers/steps thus fixed overall "effective depth". While the autoregressive "inductive scratchpad" gives the model essentially unlimited depth -- and the problem is learnable as long as each step doesn't require more than, say, 24 attention layers, the model can learn the true solution. And the model is trained to approximate BFS using a hand-designed behavior cloning expert.

But there is no mention of this (or any other) alternative explanation of the experiments, and it is assumed that the cause is globality of the global task vs locality of the individual steps. But, again, there is no experiment showing that local steps are in fact easier to learn.

There is also little mention of existing work that connects chain-of-thought reasoning to a smaller sequence of "local" steps. There are no external baselines, the evaluation is on a new "benchmark" proposed in this paper, and the approach is not evaluated on any existing benchmarks.



This brings me to my next major concern: some meaningful connection to existing work is missing.

### References:
The lack of connection to existing work makes it a little hard to tell: what is the main point of the paper? Is it globality/locality, the visual implementation of the scratchpad, or the experiments on effective depth.
1. Connecting CoT to locality: (Why think step by step? Reasoning emerges from the locality of experience. https://arxiv.org/abs/2304.03843. NeurIPS 2023 (oral))
2. Visual reasoning : (e.g. Visual Programming: Compositional visual reasoning without training (CVPR 2023 Best Paper),
ViperGPT: Visual Inference via Python Execution for Reasoning (ICCV 2023 https://viper.cs.columbia.edu/). These are both quite general -- they don't require hand-designed scratchpad structures for behavior cloning--and they for real-world tasks. There is a lot of work on visual memory/reasoning
3. The idea of effective depth has been around for a while in vision (e.g. Feedback Networks (CVPR 17) https://arxiv.org/abs/1612.09508).

**Questions:**

--Previous--
I didn't fully understand the connection between the globality degree definition and the masking experiment.
Since the BFS examples are kind of simple tasks, you could probably estimate the globality degree analytically. How do the experimental results in Fig 3 + 4 match the predicted globality degree

---

> ### Author Response · Authors · 2024-11-21
>
> **Globality vs. experiments**
>
> We kindly ask the reviewer to clarify the statement “My main concern is that the experiments don't really support the main story of the paper. They seem related at a glance, but after thinking about it some more I don't think they actually are.”.
>
> We believe that the experiments not only support the claims of the paper but they are clearly aligned with it. So we think there is a misunderstanding about the paper or about the reviewer’s comment. We add clarifications below. Hopefully, this clarifies the alignment, otherwise please provide more precise feedback. We are open to discuss and improve the writing of the paper if it happens to lead to misunderstandings.
>
> **Clarification of globality and relation to the experiments:**
> First of all, we did not define the globality as the “proportion of image patches required to predict the label”. It is more subtle: the number of patches required to get *non-trivial information* about the label. As a result, high globality does not just say that the task requires many patches to be predicted, but that one in fact needs many patches to even get some non-trivial information about the label. For instance, “counting” is not high globality, despite requiring many patches to fully predict.
>
> Below we explain why we believe the experiments align with the globality story:
>
> * *No scratchpad* models struggle with high-globality tasks because the problem remains highly global and the model does not get to learn anything meaningful. Our experiments are in agreement with this.
> * *Single-frame scratchpads* are shown to break the globality barrier and our experiments in fact show that the model can suddenly learn in these settings. It is expected that extra supervision helps with the learning, but our story explains exactly why (with the globality measure) and also shows that fairly weak scratchpads can already break the globality (with the staircase phenomenon detailed both analytically and experimentally).
> * *Inductive scratchpads* break globality like the single-frame scratchpad, again in agreement with our story and the globality measure, but have also an additional property: they are more dynamic and can length-generalize on the cycle length. I.e., they offer both in-distribution and OOD generalization in contrast to the single-frame scratchpad that only gets in-distribution generalization.
>
> The relationship between globality and the experiments is clear: tasks with high globality are hard for models to learn and generalize, necessitating the use of visual scratchpads to reduce learning complexity. Further, we specify that hardness precisely: A model of polynomial size would require exponential sample complexity (and thus training time) to learn a non-constant globality degree task.
>
> **On transformers and global attention:** Transformers, despite their global attention mechanisms, do not inherently solve high-globality tasks. This is in fact an important point: it is not sufficient to have global attention to successfully learn with high globality. For example:
>
> * In the absence of scratchpads, models fail to learn high-globality tasks like Cycles and Strings
> * The Strings task, with as few as 5 steps, cannot be solved by deep models like ViT-H, which have 32 layers making it clear that even a large number of global attention operations is insufficient. Note that this is true even when using the single-frame scratchpad.
> * The same observation has been made in the text domain. For example, it’s been shown that learning parities (a symmetric global function) using Transformers is hard. \[1,2\]
>
> 	\[1\] Theoretical limitations of self-attention in neural sequence models, Michael Hahn, 2020
> 	\[2\] Why are sensitive functions hard for transformers, Michael Hahn and Mark Rofin, 2024

---

> ### Author Response · Authors · 2024-11-21
>
> **Effective depth, alternative explanations and locality of steps**
>
> **We believe there to be no meaningful connection between the number of reasoning steps and the required minimum depth of Transformers**; we further explain this. First, note that the Cycles task of size 24 is solved with 6 steps of BFS (each small cycle is 12 nodes, expand 2 adjacent nodes at a time). Similarly, the Strings task of size 20 is solvable in 5 steps. Given Figure 5,
>
> 1. For cycles(24), without scratchpad, ViT-huge cannot solve the task while having depth 32. However, when using a single-frame scratchpad, it can solve it. Note that the expressive power of these two models is the same. So it’s the supervision of the model that makes it learn the task and not the depth of the model.
> 2. For strings(20), even ViT-huge cannot solve the task, despite having 6x layers compared to the number of reasoning steps.
>
> To gain more context on this, note that \[Abbe et al. 2024\] has proved formally that no Transformer with polynomial size (irrespective of depth) can learn the cycle task in the text domain in poly-time. No properties of effective depth will change this outcome.
>
> If the reviewer meant something different from what we argue here, we invite them to clarify. We are happy to improve our explanations in the paper if more feedback is provided.
>
> **Connection to Existing Work**
>
> We appreciate the reviewer's suggestions for related work. Below, we clarify our contributions and how they align with prior research:
>
> 1. While CoT has been explored for textual tasks (e.g., “Why Think Step by Step?”), our work adapts this to visual tasks, requiring unique considerations for 2D spatial data and pixel-wise reasoning.
> 2. Existing works like ViperGPT and Visual Programming focus on compositional reasoning or symbolic tasks, whereas we introduce interpretable benchmarks that test fundamental reasoning abilities from scratch. ViperGPT and Visual Programming papers use language models while we work in a vision-only setting.
>
> We will incorporate these references into the revised version to contextualize our contributions better.
>
>  **Benchmark**
>
> Our benchmarks are designed to address a gap in the field: existing literature often tests visual tasks relying on local features, leaving high-globality tasks unexplored. Key contributions of our benchmarks:
>
> 1. High-globality visual tasks that challenge modern vision models.
> 2. A platform to evaluate scratchpad techniques for visual reasoning.
> 3. Insights into inductive reasoning and OOD generalization in the visual setting
>
> We are not aware of any other visual benchmark that is not learnable for large Transformers with a large training set. Note that the complexities of the tasks are adjustable in our paper.
>
>  **Globality degree estimation**
>
> Estimating globality analytically for tasks like BFS is challenging due to dependencies on image resolution and patch size. Instead, we empirically demonstrate the high globality of our tasks. Models trained on ImageNet succeed with 90% masking, while even 30% masking makes Cycles unlearnable (Figure 3).
> This empirical approach avoids computing mutual information estimates while clearly showing the globality of the tasks. (Note that one has to compute the mutual information of every $k$ token with the label, given the definition. This results in a combinatorially large number of mutual information terms to be estimated.)
>
> **Response to questions**
>
> 1. The masking experiment (Figure 3\) demonstrates that ImageNet is a low-globality task, as it is solvable even with significant patch masking. In contrast, our tasks fail with masking, underscoring their high globality.
> 2. Our work shows (a) the limitations of current vision models on high-globality tasks, (b) the necessity of visual scratchpads for learning such tasks in-distribution, (c) the importance of inductive scratchpads for OOD generalization.
>
> Please let us know if any part is still unclear. We believe we have answered most of your concerns and we hope you consider increasing your score accordingly.

---

> ### Comment · Reviewer_Nq7d · 2024-11-24
>
> **TL;DR**
>
> Thanks to the authors for the technical clarifications about the experiments -- I found the discussion helpful, and I largely accept the technical points about the experimental results and globality degree. Therefore I would like to check with the authors about what they see as the main claim these experiments should support.
>
> I currently still feel that the experiments are not representative enough of computer vision tasks to substantiate the claim this is a “study of locality and globality in computer vision”. I appreciate the chance to clarify the statement, and I also appreciate the authors’ very constructive discussion. I hope the suggestions about alternative benchmarks below provide a similarly constructive discussion to the authors and reviewers.
>
>
>
> **Checking understanding about the claims of the paper: bringing [Abbe et al. 2024] to a vision setting**
>
> > We believe that the experiments not only support the claims of the paper but they are clearly aligned with it. So we think there is a misunderstanding about the paper
>
> My understanding is that the contribution of this paper is in applying the results of [Abbe et al. 2024] but now in a vision context. While [Abbe et al. 2024] introduced the theory, proofs, and showed supporting experiments; those supporting experiments are in a textual setting. This submission is mainly experimental, and shows experiments in the image domain.
>
> In particular, the experiments aim to show that for vision, too, scratchpads provides additional process supervision and learning curriculum that makes the optimization tractable for high-globality-degree tasks.
>
> Do the authors feel that characterization is a fair one?
>
> **Why I believe the experiments need to be representative of a real-world vision setting**
>
> [Abbe et al. 2024] contributed both the definition of globality degree and the formal proofs. Therefore, I believe that the additional contribution here is in extending that work to the image domain. The paper defines some of the contribuitons as “Enabling Global Reasoning in Vision” (title, L0) and as a “study of locality and globality in computer vision” (L95). To make this study more useful, I believe the study's experiments should in settings that are representative of typical computer vision.
>
>
> **Why the current experiments are not representative of a vision setting**
>
> As the authors say, they are not aware of any "visual benchmark that is not learnable for large Transformers with a large training set", hence the need for the new benchmark. I appreciate that the authors showed results using vision transformers (ViTs with inputs of shape B x H x W x C), but the benchmark images themselves are very unlike natural images used for typical computer vision
>
> - This benchmark seems like an image analogy of the benchmark of some of the experiments used in [Abbe et al 24], which “learned BFS on text tokens”. Instead of using text and a transformer decoder, these experiments use images + a transformer encoder (ViT). Essentially, these networks must “learn BFS from image patches”, and these the images are different visual depictions of graphs: node/edges, strings, mazes. While the depiction of the task is in a 2D tensors, these images are missing elements of motion blur, heavy occlusion, etc that are characteristic of real-world settings.
>
> The current benchmark doesn’t feel like a problem that requires vision or is inherently visual. But I believe there is probably some vision task requires inductive reasoning?
>
> **Vision settings that might be amenable to scratchpads**
>
> My recommendation is that the paper would be stronger with experiments that use natural images. Perhaps there are other types of image tasks that do have a high globality degree / require inductive reasoning reasoning on images.
>
> For example, the authors could consider tasks that are currently solved with RL from pixels, that have sparse reward (1/0). For example: low-level behaviors like manipulation (e.g. [Maniskill](https://github.com/haosulab/ManiSkill)) or high-level ones like in [Behavior-1K](https://arxiv.org/abs/2403.09227).
>
> Training models with sparse rewards is difficult because the signal in the search problem is very hard. The type of process supervision suggested here would reduce the problem to behavior cloning, and the specific architectural technique proposed here would be a novel approach.

---

> > ### Author Response · Authors · 2024-11-26
> >
> > We appreciate the reviewer’s engagement in the discussion.
> >
> > **Is the main contribution of the paper bringing the results of \[Abbe et al. 2024\] to a vision setting?**
> >
> > While it is true that this work extends the work of \[Abbe et al. 24\] to the visual domain, that's not the only contribution.
> >
> > * **Proposing benchmarks:** Unlike prior works on scratchpads, which operate in the text domain, our work defines and introduces high-globality benchmarks specifically tailored for vision. These benchmarks reveal that global reasoning is a critical gap in current vision models and provide controlled environments to study globality experimentally. While not all vision tasks are global, we argue that benchmarks like these are vital as we move towards generalist vision models capable of solving a broad range of tasks, including inherently global ones.
> > * **Showing that some tasks are not learnable without scratchpads in vision:**
> >   * \[Abbe et al. 2024\] formalized globality in the text domain, where scratchpads and chain-of-thought (CoT) techniques are now well-accepted. **Our work is the first to show the necessity of scratchpads in the vision domain.** Unlike text, where the need for scratchpads has been established (\[Nye et al. 2021\]), it was not anticipated that vision tasks would benefit similarly.
> >   * Vision presents unique challenges that do not exist in text, such as the difficulty in analytically computing task globality. For this reason, we experimentally demonstrate the importance of the notion of globality through carefully controlled tasks and comparisons between local and global datasets (e.g., ImageNet vs. Cycles).
> > * **Differences in scratchpad modeling:**
> >   * Vision models, unlike text models, deal with bidirectional rather than unidirectional token relationships, making autoregressive modeling inherently more challenging. In text scratchpad, the autoregressive unit is a single token, whereas in our work, it is an entire frame composed of many visual tokens. This requires adapting the transformer with multiple forward passes to achieve autoregression in vision.
> >   * Our work evaluates multiple variations of scratchpad implementations, showing their relative strengths for in-distribution (ID) and out-of-distribution (OOD) generalization.
> > * **Addressing the continuous input/output space in vision:** Vision operates in a continuous space, which introduces a distribution shift between generated scratchpad frames and ground-truth samples. To overcome this, we propose a novel training recipe that combines teacher forcing with generated frames, ensuring robust learning despite these challenges. This aspect is unique to the visual domain and has not been studied in prior works on text scratchpads.
> > * **Dynamic halting:** In text scratchpads, halting is straightforward because each token is processed sequentially until the EOS token is generated. As mentioned above, in vision this is more complicated, as autoregression is implemented via multiple forwards. Our paper introduces a dynamic way to take halting decisions based on intermediate steps, which also improves performance.
> > * **Staircase phenomenon:** As the auto-regressive unit in our work is a single frame, we observe an interesting hierarchical learning (staircase) phenomenon for each scratchpad frame. This aspect is exclusive to vision tasks and our paper uniquely identifies this behavior.
> > * **Ablations, pretrained models and scaling:** We finally note that all of the contributions above are accompanied with extensive ablations and analyses of the roles of pretraining and scaling. While pretrained models provide some benefit for simpler global tasks, our benchmarks show that without scratchpads, even these models fail on more complex tasks. This was not predicted nor experimentally validated in previous work. Moreover, we go beyond \[Abbe et al. 2024\] by presenting more detailed scaling analyses, with four model sizes and OOD results when scaling.
> >
> > Based on the evidence above, we believe it is reductive to characterize this work as a mere extension of \[Abbe et al. 2024\] or other prior text scratchpad works. In addition, we think this paper has the potential to stimulate the research community to explore visual reasoning through the lens of globality. For these reasons, we believe our paper deserves a place in the conference.

---

> > > ### Author Response · Authors · 2024-11-26
> > >
> > > **Are current datasets representative of vision tasks or inherently visual?**
> > >
> > > First note that even whether the cycle task could be considered visual or not is up for debate since deciding on connectivity of a graph is usually easier in the visual domain rather than the text domain. For example, \[Hsu et al. 23\] practically converts text graph connectivity questions into visual representations and then solves them visually. Nonetheless, increasing the visual challenges of the tasks is exactly the reason why we have created the strings and the circular maze datasets. We argue that the strings dataset is inherently visual as there is no natural symbolic/text representation for it. We would also argue that the maze datasets especially, the circular one, are visually challenging as well, and it is hard given a picture of the maze to derive a text representation of it.
> > >
> > > We agree with the reviewer that our images are not noisy, nor have blurs or heavy occlusion. However, note that our tasks are difficult for the current models even without these factors. Introducing noise or occlusion can only make our tasks harder. We have opted for the simplest version of the tasks to avoid introducing confounding factors into our analysis. Nevertheless, our arguments and methodology will remain valid for more complex versions of the proposed tasks whether it's a more complex process for generating the mazes or different distortions added to our images.
> > >
> > > We agree with the reviewer that the claim in the sentence “a study of locality and globality in computer vision” needs to be smoothed down. We will update the pdf to reflect this. Thanks for the effort in improving the paper.
> > >
> > > \[Hsu et al. 23\] "Can Visual Scratchpads With Diagrammatic Abstractions Augment LLM Reasoning?, Hsu et. al. NeurIPS 2023"
> > >
> > > **Extending this work to other tasks** **such as Maniskill or Behavior-1k**
> > >
> > > We thank the reviewer for suggesting these datasets. We indeed agree that our methods are suitable for learning from videos of such environments. Nevertheless, note that these tasks are rather different from what we have studied here. The tasks appearing in Maniskill/Behavior-1k have state transitions and require several steps to be solved and are thus *inherently sequential*. What we have shown here is that there are image classification tasks (hence, the task itself is not sequential, e.g., doesn’t have a sequential output) for which we have to break down the task and use an auto-regressive visual scratchpad. We believe the use of synthetic datasets in this work (with well-defined generation process/distribution) has allowed us to focus on the core issue of globality, show how severe globality can impact learning and the necessity of scratchpads, and provide an understanding of the whole picture. Note that even the original scratchpad paper \[Nye et al. 21\] in the text domain considered mostly synthetic tasks; and gathering suitable image/video data could be much more challenging than text data.
> > >
> > > We think the reviewer is right in advocating for more visual benchmarks that incorporate global reasoning challenges. In part, we have already taken steps in this direction by proposing visually challenging versions of connectivity and solvability tasks, such as the Strings and Circular Maze datasets. These tasks are designed to require visual reasoning because textual or analytical representations of the problems are impractical or non-existent (e.g., there is no way to write a symbolic representation of the Strings task).
> > >
> > > However, extending our methodology to more "natural image" settings involves significant challenges, such as the need for generative models or extensive data collection. These tasks are complex enough to warrant their own dedicated studies and papers.
> > >
> > > Additionally, we have already included extensive results, analyses, and experiments, as shown by the rich appendix, and this project has consumed around 160k GPU hours on high-end hardware like A100 and H100 GPUs. Further expansion would go beyond the scope of this project.
> > >
> > > As a testimony to our alignment with the reviewer’s vision, we are actively exploring scratchpads in more realistic and complex settings, such as solving geometric problems, circuit analysis, map understanding, and other tasks requiring intricate reasoning like games and puzzles. These directions require separate data collection and preparation efforts and multiple modalities to work together. While related, such work deserves much more extensive exploration than what we can provide in this rebuttal.

---

> > > > ### Comment · Reviewer_Nq7d · 2024-11-27
> > > >
> > > > **Final decision**
> > > >
> > > > I am raising my score from 3 ->(somewhere between a 4 and a 5), because after a discussion with the authors I don't see significant technical or methodological errors. I can't comment on whether the method is novel or not -- I'm not familiar enough with the methods that mentioned by reviewer 83P1 to say one way or the other.
> > > >
> > > > The score is a 5, not 6, because all the experiments in the current paper are some form of cycle detection on a graph. While I believe the cycles task is a valid benchmark and good for demonstrating the theory and approach, the cycles task itself is not very representative of most computer vision tasks like recognition, reconstruction, detection, or visual navigation/manipulation. The paper is written in a way that suggests the method is supposed to be used for computer vision tasks.
> > > >
> > > > The cycles task _does_ resemble some the other (important) settings the authors mentioned they are actively exploring -- and I would love to see the results of the authors current exploration of "scratchpads in more realistic and complex settings, such as solving geometric problems, circuit analysis, map understanding, and other tasks requiring intricate reasoning like games and puzzles."
> > > >
> > > > If I reviewed that paper instead of this one, I would recommend acceptance -- but I feel the current version would require a major revision (either in writing or in experiments). While at this point I basically like the paper, I am not comfortable recommending acceptance for a paper that I believe requires any sort of major revision.
> > > >
> > > > The authors are invited to provide their final thoughts or responses to this comment if they wish.

---

> > > > > ### Author Response · Authors · 2024-12-01
> > > > >
> > > > > We sincerely thank the reviewer for their openness to discussion, their constructive feedback, and their willingness to reassess the score after our exchanges.
> > > > >
> > > > > It was never our intention to mislead readers into thinking that the proposed global benchmarks are representative of the full spectrum of computer vision tasks. Instead, our work focuses on analyzing certain cases of truly global tasks, explicitly designed to avoid spurious correlations between local features and labels and require multi-step reasoning. That said, we acknowledge that the two points raised by the reviewer (the title and line 95) may have inadvertently led to misunderstandings. These have been revised to clarify our intent, and we are happy to address any additional concerns the reviewer identifies.
> > > > >
> > > > > Regarding novelty and differences from prior work, our contributions are distinguished by both objective and methodology, please see our response to reviewer 83P1.
> > > > >
> > > > > With this context in mind, we respectfully disagree that the paper requires a major revision. We believe the paper in its current state does a good job of developing the theory of globality for vision, proposing global tasks that are challenging both theoretically and experimentally, demonstrating their fundamental differences from mainstream computer vision tasks like ImageNet, and introducing methods to address these tasks and generalize to similar but harder versions. We have further completed the paper with extensive ablation analyses. While we agree that this work does not exhaustively address globality in vision, we believe it provides a meaningful and rigorous first step in this direction. As mentioned earlier, this project involved significant effort and has already yielded many valuable findings (see above). While all papers can be improved, we believe the current version is worthy of being shared with the community.
> > > > >
> > > > > We appreciate the reviewer’s feedback and encouragement and would be glad to provide any additional clarifications if needed.

---

### Official Review · Reviewer_fCz5 · 2024-11-06

**Soundness:** 3
**Presentation:** 3
**Contribution:** 3
**Rating:** 8
**Confidence:** 4

**Summary:**

The paper deals with visual problems that require global reasoning. The proposed tasks are based on the connectivity problems discussed by Minsky and Papert in 1969. The paper shows that large vision models of today still struggle with learning efficiency when dealing with visual problems that require global reasoning. To deal with this issue the paper introduces a "visual scratchpad" based on text scratchpads and chain-of-thoughts used in language models.

**Strengths:**

* The paper identifies and address an important problem. The paper introduces the novel Cycles and Strings tasks which turn out to be challenging for current large vision models.

* The paper provides a good theoretical analysis of tasks that require global reasoning through the definition of globality degree.

* The paper includes extensive experiments that show that current large vision models cannot deal with global reasoning problems irrespective of model size (Figure 5).

**Weaknesses:**

* Novelty of Scratch Pads: Visual scratch pads have already been explored in "Can Visual Scratchpads With Diagrammatic
Abstractions Augment LLM Reasoning?, Hsu et. al. NeurIPS 2023".

* Implementation details: Some very important details are not clear -- in L335 "add a linear layer to the hidden representation of the last transformer layer to predict the scratchpad image". Use of a simple linear layer would likely severely limit the resolution of the output scratch pad image. It would be useful if the paper discusses the resolution limits (if any) of the visual scratch pad, as this would limit the complexity of the problems that can be tacked by the proposed approach.

* Error propagation: For complex tasks, without a sophisticated visual scratch pad generation mechanism, pixel level errors might have a significant impact on reasoning capabilities.

* Baselines: The paper does not consider state of the art VLMs such as LLaVA or InstructBLIP as baselines. As these models use more sophisticated attention mechanisms, it is possible that the proposed Cycles and Strings tasks can be solved by such models.

* Compute Cost: The use of visual scratch pads would add a significant compute overhead. This should be discussed in more detail.

**Questions:**

* The novelty of the proposed approach should be discussed in more detail.
* The implementation details of the visual scratch pads should be discussed in more details.

---

> ### Author Response · Authors · 2024-11-21
>
> **Novelty of scratchpads:** The concept of visual scratchpads has some resemblance to the work by Hsu et al. (2023), and we will cite it in the revised manuscript. However, there are crucial differences in scope and implementation:
>
> * **Domain:** Hsu et al. focus on augmenting language model reasoning through external tools that generate visual diagrams for text-based tasks. In contrast, our work centers on vision-only tasks requiring global reasoning, such as solving mazes and analyzing graph connectivity (we target hard visual tasks in contrast to this work.)
> * **Scratchpad implementation:** In Hsu et al., the visual diagram is created externally by tools. Conversely, in our framework, the model itself generates the scratchpad, breaking down global problems into intermediate visual reasoning steps inside the model. This includes the use of autoregressive, multi-frame predictions, a novel approach not explored in Hsu et al.
> * **Task globality:** The tasks we introduce (Cycles, Strings, and Mazes) are different in nature, as they are high-globality visual benchmarks, unlike the textual reasoning tasks in Hsu et al. To the best of our knowledge, there are no similar visual benchmarks in the literature.
>
> **Implementation details (linear layer for scratchpad):** The linear layer operates on **each patch representation**, not the CLS token. This ensures that the model retains the spatial resolution of the input image. As shown in Figure 7 and Appendix F, the quality of the generated scratchpads is sufficient to solve the proposed tasks. Also, while our framework uses a simple linear layer for its output, it is not constrained to this choice. More sophisticated generation mechanisms can be integrated into our method if necessary. However, we opted for simplicity to avoid introducing confounding factors.
>
> **Error propagation:** Error propagation can be a concern. To mitigate this, we adopt **alternated training** (described in Section 3.2 and Appendix C.1), where the model is exposed to both perfect training frames and generated frames. This strategy is effective in reducing the gap between training and inference performance, as experimentally validated in the ablation study in Section 4.3.
>
> **Baselines:** Our focus is on analyzing and understanding **global vision-only reasoning tasks from scratch**, not benchmarking against state-of-the-art Vision-Language Models (VLMs). However, we agree that it is interesting to check what the current state-of-the-art is capable of. Preliminary tests with VLMs like LLaVA and InstructBLIP (via image input and text prompting) show that these models struggle with our high-globality tasks. Interestingly, the answer seems to be more correlated with the given prompt rather than with the image. Moreover, even more complex state-of-the-art systems like ChatGPT, Claude and Gemini are unable to tell solvable vs non-solvable mazes apart. The reviewer can test that on their own by taking a screenshot of the figures in the paper and pasting them in their chatbot of choice.
>
> **Compute cost:** While visual scratchpads introduce some overhead at inference (due to multiple forward passes for multi-frame scratchpads), this is offset by the ability to use **smaller models**. For instance, the inductive scratchpad achieves comparable performance with fewer parameters (see Figure 5). Additionally, the compute cost is adaptive, scaling with task complexity, which is a desirable property, rather than a weakness.
>
> **Response to Questions:**
>
> **Novelty discussion:** The novelty of our paper is twofold:
>
> * We propose a theoretical framework to analyse the globality / locality of tasks and we validate it empirically.
> * We apply the scratchpad concept to visual tasks requiring global reasoning. We introduce both single-frame and multi-frame (inductive) scratchpads and demonstrate their effectiveness on novel, challenging datasets. We also analyse in domain and out-of-domain performance of each method.
>
> **Implementation details:**
>
> * As mentioned, the scratchpad prediction head uses a linear layer applied to patch embeddings. The spatial resolution of the output is a function of the patch resolution and sequence length. Therefore the fact that the prediction head is not very powerful is not an issue, as it only needs to map a patch representation to a few pixels (16x16). Moreover, empirically we found this not to be problematic, examples of generated scratchpads, illustrating their quality, are provided in Figure 7 and Appendix F.
>
> Please let us know if any part is still unclear. We believe we have answered most of your concerns and we hope you consider increasing your score accordingly.

---

> > ### Comment · Reviewer_fCz5 · 2024-11-26
> > **Concerns Addressed**
> >
> > The rebuttal has addressed my concerns. I will raise my score.

---

### Author Response · Authors · 2024-11-28
**Paper revision (new PDF uploaded)**

We thank the reviewers for their suggestions. Accordingly, we have improved the paper and uploaded a new version on OpenReview. Changelog:
- Title: “Visual scratchpads: enabling global reasoning in vision” → “Visual scratchpads: enabling global visual reasoning”
- Line 95: “Study of locality and globality in computer vision” → “Exploration of locality and globality in the visual domain”
- Minor rewording:
  - In Ablations (Section 4.3)
  - In the Introduction, paragraph starting at line 054

As the discussion with reviewers regarding prior work is ongoing, we will further improve the related work section once the discussion concludes. We will ensure all relevant prior work is appropriately incorporated.

---

### Meta-Review · Area_Chair_tX6U · 2024-12-19

**Metareview:**

The paper explores visual reasoning through synthetic connectivity tasks with significant methodological limitations. The proposed visual scratchpad approach lacks novelty, as prior works have developed similar sequence modeling techniques. The experimental design is narrow, focusing on cycle-based tasks that do not represent real-world visual reasoning challenges. The authors' claims about globality remain ambiguous, with experimental validation restricted to constrained synthetic tasks. The supervision mechanism appears to drive performance improvements more than the scratchpad technique, raising concerns about generalizability. The lack of diverse baselines and limited out-of-distribution testing further weaken the paper's contributions.

**Additional Comments On Reviewer Discussion:**

Reviewers critiqued the paper's narrow experimental scope and lack of novelty, highlighting that the connectivity tasks did not represent real-world visual reasoning. Authors attempted to defend their approach by explaining the globality concept and comparing their method to prior work. However, reviewers remained unconvinced, maintaining that the experiments were too limited and the supervision mechanism appeared to drive performance more than the proposed technique. Despite extensive rebuttals, the fundamental limitations in experimental design and practical applicability persisted.

---

### Decision · Program_Chairs · 2025-01-22

Reject